# A transcriptomic atlas of mouse cerebellar cortex comprehensively defines cell types

Velina Kozareva[1,4], Caroline Martin[1,4], Tomas Osorno[2], Stephanie Rudolph[2], Chong Guo[2], Charles Vanderburg[1], Naeem Nadaf[1], Aviv Regev[1], Wade G. Regehr[2] & Evan Macosko[1,3✉]

The cerebellar cortex is a well-studied brain structure with diverse roles in motor learning, coordination, cognition and autonomic regulation. However, a complete inventory of cerebellar cell types is currently lacking. Here, using recent advances in high-throughput transcriptional profiling[1–3], we molecularly define cell types across individual lobules of the adult mouse cerebellum. Purkinje neurons showed considerable regional specialization, with the greatest diversity occurring in the posterior lobules. For several types of cerebellar interneuron, the molecular variation within each type was more continuous, rather than discrete. In particular, for the unipolar brush cells—an interneuron population previously subdivided into discrete populations—the continuous variation in gene expression was associated with a graded continuum of electrophysiological properties. Notably, we found that molecular layer interneurons were composed of two molecularly and functionally distinct types. Both types show a continuum of morphological variation through the thickness of the molecular layer, but electrophysiological recordings revealed marked differences between the two types in spontaneous firing, excitability and electrical coupling. Together, these findings provide a comprehensive cellular atlas of the cerebellar cortex, and outline a methodological and conceptual framework for the integration of molecular, morphological and physiological ontologies for defining brain cell types.

The cerebellar cortex is composed of the same basic circuit replicated thousands of times. Mossy fibres from many brain regions excite granule cells that in turn excite Purkinje cells (PCs), the sole outputs of the cerebellar cortex. Powerful climbing fibre synapses, which originate in the inferior olive, excite PCs and regulate synaptic plasticity. Additional circuit elements include inhibitory interneurons such as molecular layer interneurons (MLIs), Purkinje layer interneurons (PLIs), Golgi cells, excitatory unipolar brush cells (UBCs) and supportive Bergmann glia. There is a growing recognition that cerebellar circuits exhibit regional specializations, such as a higher density of UBCs or more prevalent PC feedback to granule cells in some lobules. Molecular variation across regions has also been identified, such as the parasagittal banding pattern of alternating PCs with high and low levels of *Aldoc* expression[4]. However, the extent to which cells are molecularly specialized in different regions is poorly understood.

Achieving a comprehensive survey of cell types in the cerebellum poses some unique challenges. First, a large majority of the neurons are granule cells, making it difficult to accurately sample the rarer types. Second, for many of the morphologically and physiologically defined cell types—especially the interneuron populations—existing molecular characterization is extremely limited. Recent advances in single-cell RNA sequencing (scRNA-seq) technology[1–3] have increased the throughput of profiling to enable the systematic identification of cell types and states throughout the central nervous system[5–8]. Several

recent studies have harnessed such techniques to examine some cell types in the developing mouse cerebellum[9–11], but none has yet comprehensively defined mature cell types in the adult.

## Identification of cerebellar cell types

We developed a pipeline for high-throughput single-nucleus RNA-seq (snRNA-seq) with high transcript capture efficiency and nuclei yield, as well as consistent performance across regions of the adult mouse and post mortem human brain[12] (https://doi.org/10.17504/protocols.io.bck6iuze; Methods). To comprehensively sample cell types in the mouse cerebellum, we dissected and isolated nuclei from 16 different lobules, across both female and male replicates (Fig. 1a, Extended Data Fig. 1a, Methods). We recovered 780,553 nuclei profiles with a median transcript capture of 2,862 unique molecular identifiers (UMIs) per profile (Extended Data Fig. 1b, c), including 530,063 profiles from male donors, and 250,490 profiles from female donors, with minimal inter-individual batch effects (Extended Data Fig. 1d, e).

To discover cell types, we used a previously developed clustering strategy[12] (Methods) to partition 611,034 high-quality profiles into 46 clusters. We estimate that with this number of profiles, we can expect to sample even extremely rare cell types (prevalence of 0.15%) with a probability of greater than 90%, which suggests that we captured most transcriptional variation within the cerebellum (Extended Data Fig. 1f).

[1]Broad Institute of Harvard and MIT, Stanley Center for Psychiatric Research, Cambridge, MA, USA. [2]Department of Neurobiology, Harvard Medical School, Boston, MA, USA. [3]Department of Psychiatry, Massachusetts General Hospital, Boston, MA, USA. [4]These authors contributed equally: Velina Kozareva, Caroline Martin. ✉e-mail: emacosko@broadinstitute.org

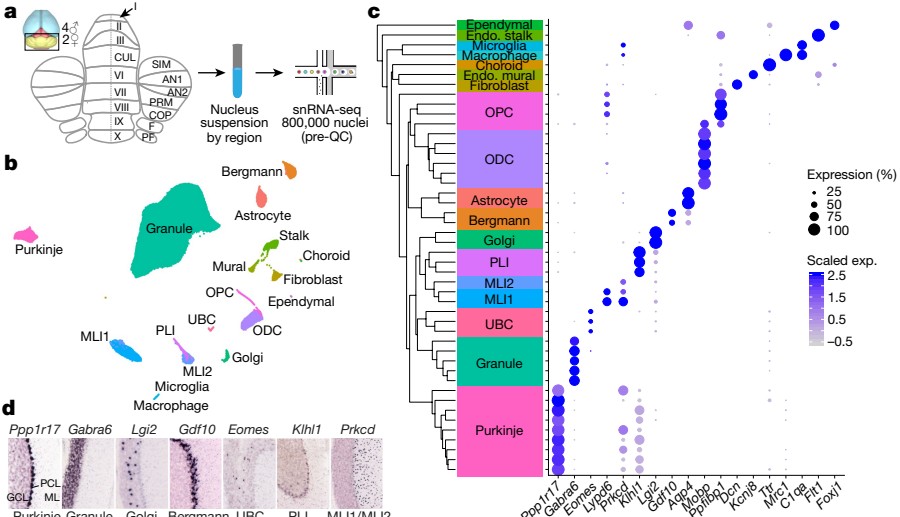

**Fig. 1 | Comprehensive transcriptional profiling of cell types across the mouse cerebellum. a**, Experimental design with lobe-based sampling and profiling. AN, ansiform lobule; COP, copula pyramidis; CUL, culmen; F, flocculus; PF, paraflocculus; PRM, paramedian lobule; SIM, simple lobule. **b**, UMAP visualization of 611,034 nuclei (after profile quality control and annotation; Methods), coloured by cell type identity. ODC, oligodendrocyte; OPC, oligodendrocyte precursor cell. **c**, Dendrogram indicating hierarchical relationships between cell subtypes (left) with a paired dot plot (right) of scaled expression (exp.) of selected marker genes for cell type identity. Background text colours correspond to cell types in **b**. **d**, Allen Brain Atlas expression staining for selected gene markers of canonical cell populations, indicating cerebellar layer localization. GCL, granule cell layer; ML, molecular layer; PCL, Purkinje cell layer.

We assigned each cluster to one of 18 known cell type identities on the basis of the expression of specific molecular markers that are known to correlate with defining morphological, histological and/or functional features (Fig. 1b, c, Supplementary Table 1). These annotations were also corroborated by the expected layer-specific localizations of marker genes in the Allen Brain Atlas (ABA)[13] (https://mouse.brain-map.org) (Fig. 1d). Several cell types contained multiple clusters defined by differentially expressed markers, which suggests further heterogeneity within those populations (Extended Data Fig. 2a–n, Supplementary Table 2).

## Cell type variation across lobules

To quantify the regional specialization of cell types, we examined how our clusters distributed proportionally across each lobule. We found that eight of our nine PC clusters, as well as several granule cell clusters and one Bergmann glial cluster, showed the most significantly divergent lobule compositions (Pearson's chi-squared test, false discovery rate (FDR) < 0.001; Methods) and exhibited greater than twofold enrichment in at least one lobule (Fig. 2a). There was high concordance in the regional composition of each of these types across replicates, which indicates consistent spatial enrichment patterns (Extended Data Fig. 3a).

The nine PC clusters could be divided into two main groups on the basis of their expression of *Aldoc*, which defines parasagittal striping of Purkinje neurons across the cerebellum[4]. Seven of the nine PC clusters were *Aldoc*-positive, indicating greater specialization in this population compared with the *Aldoc*-negative PCs. Combinatorial expression of *Aldoc* and at least one subtype-specific marker fully identified the Purkinje clusters (Fig. 2b). These *Aldoc*-positive and *Aldoc*-negative groups showed a regional enrichment pattern that was consistent with the known paths of parasagittal stripes across individual lobules (Fig. 2c). When characterizing the spatial variation of the PC subtypes, we found some with spatial patterns that were recently identified using Slide-seq technology (Aldoc_1 and Aldoc_7, marked by *Gpr176* and *Tox2*, respectively)[14], as well as several undescribed subtypes and patterns (Fig. 2b, d, Extended Data Fig. 3b). Most of this PC diversity was concentrated in the posterior cerebellum, particularly the uvula and nodulus, consistent with these regions showing greater diversity in both function and connectivity[15,16].

We also observed regional specialization in excitatory interneurons and Bergmann glia. Among the five granule cell subtypes (Fig. 2e), three displayed cohesive spatial enrichment patterns (subtypes 1, 2 and 3) (Fig. 2f, Extended Data Fig. 3c). In addition, and consistent with previous work[17], the UBCs as a whole were highly enriched in the posterior lobules (Extended Data Fig. 3d). Finally, we identified a Bergmann glial subtype that expressed the marker genes *Mybpc1*[14] and *Wif1* (Fig. 2g), with high enrichment in lobule VI, the uvula and nodulus (Fig. 2h, Extended Data Fig. 3e). The regional specialization of interneuron and glial populations is in contrast to the cerebral cortex, where molecular heterogeneity across regions is largely limited to projection neurons[5,7].

## Continuous variation within cell types

Molecularly defined cell populations can be highly discrete—such as the distinctions between chandelier and basket interneuron types in the cerebral cortex[6]—or they can vary more continuously, such as the cross-regional differences among principal cells of the striatum[7,18] and cortex[5,7]. The cerebellum is known to contain several canonical cell types that exist as morphological and functional continua, such as the basket and stellate interneurons of the molecular layer[19]. To examine continuous features of molecular variation in greater detail within interneuron types, we created a metric to quantify and visualize the continuity of gene expression between two cell clusters. In brief, we fit a logistic curve for differentially expressed genes along the dominant expression trajectory[20], extracting the maximum slope ($m$) of the curve (Methods, Fig. 3a). We expect $m$ values to be smaller for genes that are representative of more continuous expression variation (Fig. 3a).

Our cluster analysis initially identified three populations of UBCs, similar in number to the two to four discrete types suggested by previous immunohistochemistry studies[21–24]. However, comparing $m$ values across 200 highly variable genes within the UBC, Golgi cell and MLI populations suggested that in UBCs, many genes showed continuous variation (Fig. 3b), including *Grm1*, *Plcb4*, *Calb2* and *Plcb1* (Fig. 3c, d).

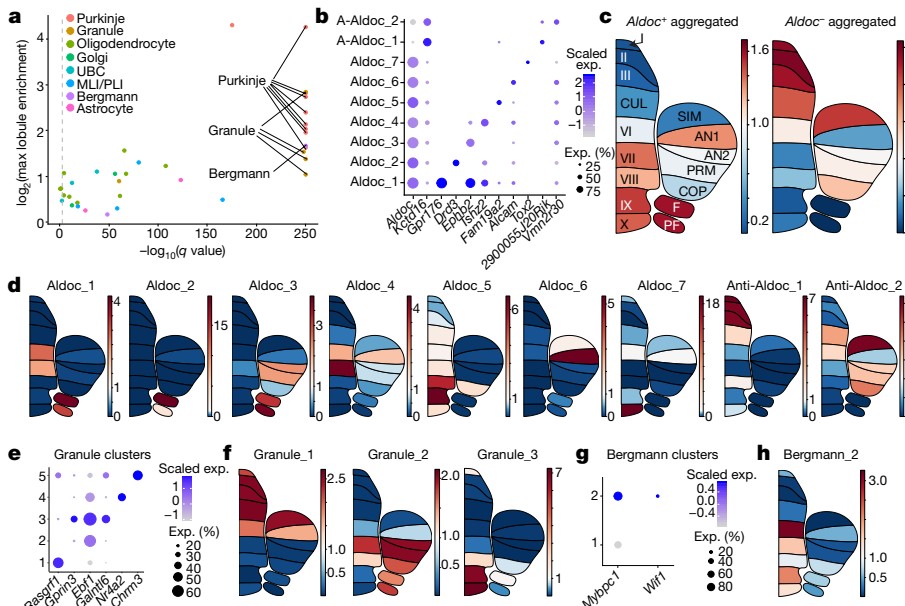

**Fig. 2 | Characterization of spatial variation and patterning in neuronal and glial cell types. a**, Plot indicating neuronal and glial clusters that have lobule enrichment patterns that are significantly different from the cell type population as a whole (Pearson's chi-squared, FDR < 0.001 indicated by dashed line). *x*-axis shows $-\log_{10}$-transformed *q* values; *y* axis shows $\log_2$-transformed maximum lobule enrichment across all 16 lobules (Methods). Clusters with high correlation (Pearson correlation coefficient > 0.85) in lobule enrichment values between replicate sets and maximum lobule enrichment > 2 are labelled (Methods, Extended Data Fig. 3a). **b**, Dot plot of scaled expression of selected gene markers for PC clusters. **c**, Regional enrichment plots indicating average

lobule enrichment for aggregated *Aldoc*-positive PC subtypes (left) and aggregated *Aldoc*-negative subtypes (right). Regions labelled as in Fig. 1a. A-Aldoc, anti-Aldoc. **d**, Regional enrichment plots indicating lobule enrichment for PC clusters. **e**, Dot plot of scaled expression of selected gene markers for granule cell clusters. **f**, Regional enrichment plots indicating lobule enrichment for three spatially significant granule cell clusters. **g**, Dot plot of scaled expression of selected gene markers for Bergman glial clusters. **h**, Regional enrichment plot indicating lobule enrichment for the *Mybpc1*-positive *Wif1*-positive Bergmann glial cluster.

Cross-species, integrative analysis[12] with cerebellar cells derived from two post mortem human donors (Methods) revealed evolutionary conservation of the continuum (Extended Data Fig. 4a), with graded expression of many of the same genes, including *Grm1* and *Grm2* (Extended Data Fig. 4b).

Functionally, UBCs have been classified on the basis of their response to mossy fibre activation. Discrete ON and OFF categories have previously been emphasized, although some properties of UBCs do not readily conform to these distinct categories[21,25,26]. Here we focused on whether the molecular gradients in the expression of metabotropic receptors readily translated to a continuum of functional properties. We pressure-applied glutamate and measured the spiking responses of UBCs with on-cell recordings, and then measured glutamate-evoked currents in the cell (Methods). In some cells, glutamate rapidly and transiently increased spiking and evoked a long-lasting inward current (Fig. 3e, top left). For other cells, glutamate transiently suppressed spontaneous firing and evoked an outward current (Fig. 3e, bottom left). Many UBCs, however, had more complex, mixed responses to glutamate; we refer to these as 'biphasic' cells. In one cell, for example, glutamate evoked a delayed increase in firing, caused by an initial outward current followed by a longer lasting inward current (Fig. 3e, middle left). A summary of the glutamate-evoked currents (Fig. 3e, right) suggests that the graded nature of the molecular properties of UBCs may lead to graded electrical response properties. To link the functional and molecular continua more directly, we recorded from cells treated with agonists of mGluR1 (*Grm1*) or mGluR2 (*Grm2*) (Fig. 3f). Responses were graded across the UBC population, with a significant number of cells that responded to both agonists (Fig. 3f, Extended Data Fig. 5). This suggests that the biphasic response profile probably corresponds to the molecular continuum defined by snRNA-seq. Further studies are needed to determine the relationship between these

diverse responses to applied agonists, and the responses of the cells to mossy fibre activation.

## Two discrete MLI subtypes

MLIs are spontaneously active interneurons that inhibit PCs as well as other MLIs. MLIs are canonically subdivided into stellate cells located in the outer third of the molecular layer, and basket cells located in the inner third of the molecular layer that synapse onto PC somata and form specialized contacts known as pinceaus, which ephaptically inhibit PCs. Many MLIs, particularly those in the middle of the molecular layer, share morphological features with both basket and stellate cells[19]. Thus, MLIs are thought to represent a single functional and morphological continuum.

Our clustering analysis of MLIs and PLIs, by contrast, identified two discrete populations of MLIs. The first population, 'MLI1', uniformly expressed *Lypd6*, *Sorcs3* and *Ptprk* (Figs. 1c, 4a). The second population, 'MLI2', was highly molecularly distinct from MLI1, and expressed numerous markers that are also found in PLIs, such as *Nxph1* and *Cdh22* (Fig. 4a). Single-molecule fluorescence in situ hybridization (smFISH) experiments with *Sorcs3* and *Nxph1* showed that the markers were entirely mutually exclusive (Fig. 4b, c). A cross-species analysis with 14,971 human MLI and PLI profiles demonstrated that the MLI1 and MLI2 distinction is evolutionarily conserved (Extended Data Fig. 4c, d).

To examine the developmental specification of these two populations, we clustered 79,373 total nuclei from peri- and postnatal mice across several time points (ranging from embryonic day (E) 18 to postnatal day (P) 16). From a cluster of 5,519 GABA (γ-aminobutyric acid)-producing neuron progenitors, marked by the expression of canonical markers *Tfap2b*, *Ascl1* and *Pax2*[27,28] (Methods, Extended Data Fig. 6a, b), we were able to distinguish developmental trajectories that

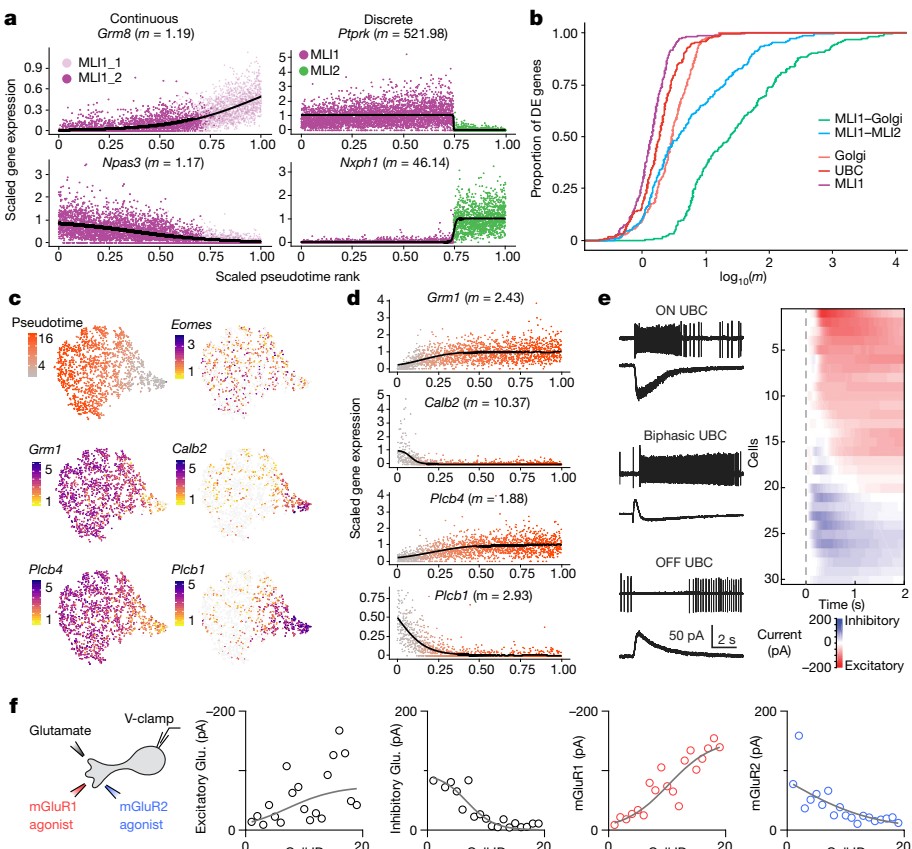

**Fig. 3 | Cross-cluster continuity among select neuronal populations, including unipolar brush cells. a**, Gene expression ordered by pseudotime (Methods) for two of the top differentially expressed genes between two clusters within the MLI1 type (left, MLI1_1 and MLI1_2), and between two cell types (right, MLI1 and MLI2 cell types). Curves represent logistic fits estimated via nonlinear least squares; maximum slope values (*m*) are indicated. Differences in magnitude of *m* values correspond well with visually distinctive molecular continuity versus discreteness. **b**, Empirical cumulative distributions of $\log_{10}(m)$ values (curve fit as in **a**) for top differentially expressed (DE) genes among aggregated combinations of MLI, Golgi cells and UBC clusters. **c**, *t*-distributed stochastic neighbour embedding (*t*-SNE) visualizations of UBCs (*n* = 1,613), coloured by pseudotime loading (top left), and log-normalized expression of canonical gene markers for all UBCs (*Eomes*), ON UBCs (*Grm1*, *Plcb4*), and OFF UBCs (*Calb2*, *Plcb1*). **d**, Pseudotime rank

ordered gene expression for canonical markers associated with ON UBCs (*Grm1*, *Plcb4*) and OFF UBCs (*Calb2*, *Plcb1*). Cells are coloured by pseudotime loading. Curves indicate logistic fits estimated as in **a**; maximum slope values (*m*) are indicated. **e**, Left, cell-attached recordings of spiking responses and whole-cell recordings of currents evoked by brief glutamate puffs in ON (top), biphasic (middle) and OFF (bottom) UBCs. Right, heat map of the currents recorded in all UBCs, sorted by the magnitude and time course of charge transfer. **f**, Left, schematic describing whole-cell recordings obtained from UBCs evoked by pressure application of glutamate (left traces, black), the mGluR1 agonist DHPG (middle trace, red) and the mGluR2 agonist LY354740 (right trace, blue) using three pipettes placed within 20 μm of the recorded cell. Evoked currents from each application are summarized in the correspondingly coloured plots. Representative UBC recordings are shown in Extended Data Fig. 5.

corresponded to the MLI1 (*Sorcs3*-positive) and MLI2 (*Nxph1*-positive, *Klhl1*-negative) populations, with differentiation of the two types beginning at P4 and largely complete by P16 (Fig. 4d, e, Extended Data Fig. 6a). Although both populations originate from a single group of progenitors, trajectory analysis revealed several lineage-specific markers (Extended Data Fig. 6c–e). Among the MLI2 trajectory markers, we identified genes such as *Fam135b*, the expression of which persisted into adulthood, and *Fos*, which is only transiently differentially expressed between the MLI1 and MLI2 trajectories (Fig. 4e, Extended Data Fig. 6e, f). This high expression of several immediate early genes (Extended Data Fig. 6f) selectively in early MLI2 cells could indicate that differential activity is associated with MLI2 specification.

MLI1s and MLI2s were present throughout the entire molecular layer, which indicates that the distinction between MLI1 and MLI2 does not correspond to the canonical basket and stellate distinction (Extended Data Fig. 7a). To understand the morphological, physiological and molecular characteristics of the MLI populations better, we developed a pipeline to record from individual MLIs in brain slices, image their

morphologies, and then ascertain their molecular MLI1 and MLI2 identities by smFISH (Methods, Fig. 4f). Consistent with the marker analysis (Fig. 4a), MLI1s had a stellate morphology in the distal third of the molecular layer, whereas MLI1s located near the PC layer had a basket morphology, with contacts near PC initial segments (Fig. 4f, Extended Data Fig. 7b). We next examined whether MLI2s, in which we could not identify systematic molecular heterogeneity, had graded morphological properties. MLI2s in the distal third of the molecular layer also had stellate cell morphology, whereas MLI2s near the PC layer had a distinct morphology and appeared to form synapses preferentially near the PC layer (Extended Data Fig. 7b). Although further studies are needed to determine whether MLI2s form pinceaus, it is clear that both MLI1 and MLI2 showed a similar continuum in their morphological properties.

The electrical characteristics of MLI1s and MLI2s also showed numerous distinctions. The average spontaneous firing rate was significantly higher for MLI1s than for MLI2s (Mann–Whitney test, *P* = 0.0015) (Fig. 4g), and the membrane resistance ($R_m$) of MLI1s was lower than that of MLI2s (Fig. 4g). In addition, we found that MLI2s

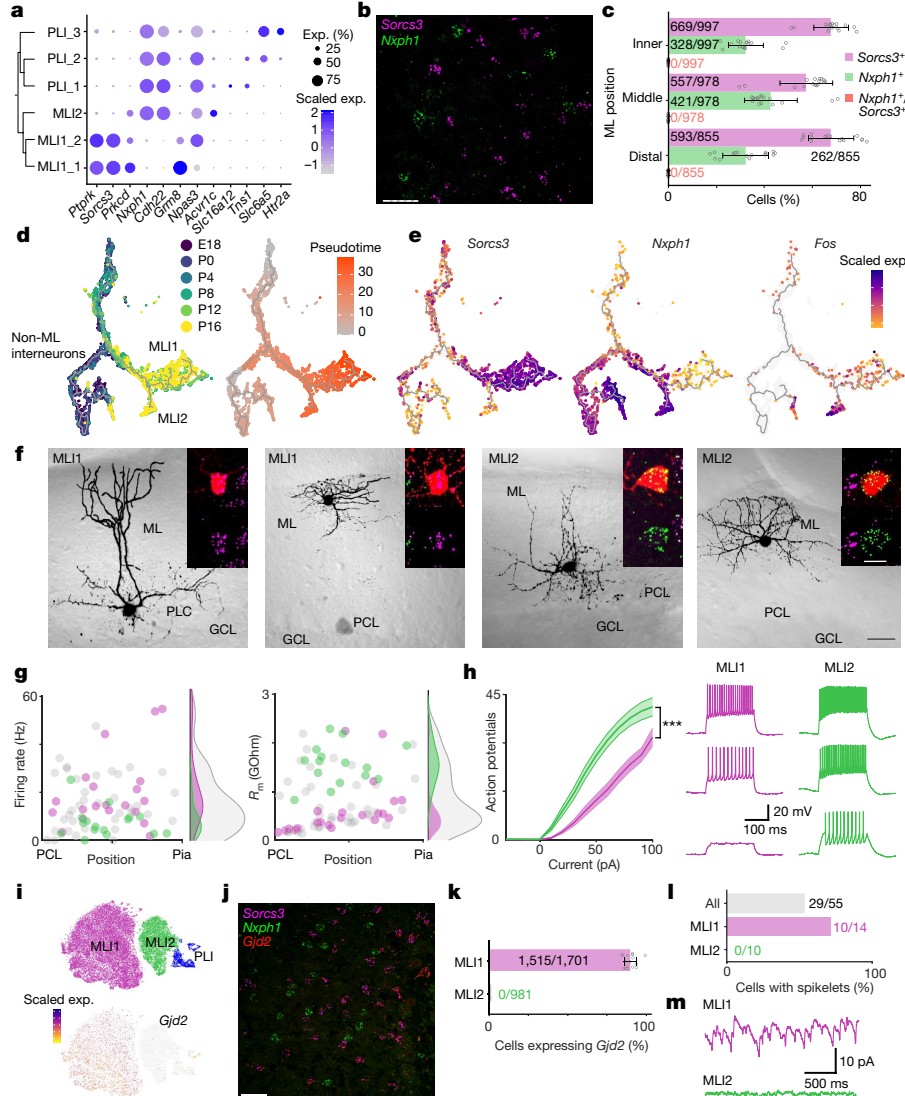

**Fig. 4 | Two molecularly and electrophysiologically distinct populations of MLIs. a**, Dendrogram of gene expression relationships (computed as in Fig. 1c) among MLI and PLI populations (left), paired with dot plot (right) of selected gene markers. **b**, Representative image of smFISH expression of *Sorcs3* (purple) and *Nxph1* (green) within the molecular layer (*n* = 16 slides sectioned from 3 mice). Scale bar, 20 μm. **c**, The percentages of *Sorcs3*+, *Nxph1*+ and *Sorcs3*+*Nxph1*+ cells in the inner third, middle third and distal third of the molecular layer (ML) (total cells counted shown; *n* = 16 slides sectioned from 3 mice). Data are mean ± s.d. **d**, UMAP visualization of cerebellar interneuron developmental trajectory, coloured by developmental age of collection (left) and by pseudotime (right) (Methods). **e**, UMAP visualization of *Sorcs3*, *Nxph1* and *Fos* expression across MLI development. **f**, Four two-photon images of representative basket and stellate-like MLI1 and MLI2 neurons in cerebellar slice. Insets show the fluorescent fill of the cell body (red) with the smFISH signals superimposed (top), and the smFISH signal only (bottom; *Sorcs3*, purple; *Nxph1*, green). Scale bars, 30 μm (black) and 10 μm (white). *n* = 23 MLI1 cells, 20 MLI2 cells. **g**, Scatter plots of firing rate (left) and membrane resistance *R*m (right) of molecularly identified MLI1 (purple, *n* = 23 cells) and

MLI2 neurons (green, *n* = 20 cells), as well as MLIs whose molecular identity was not ascertained (grey, *n* = 55 cells). Corresponding distributions at the right. **h**, Left, Mean input–output curves of MLI1 (purple) and MLI2 neurons (green). Shaded area denotes s.e.m. *n* = 22 MLI1 cells, 20 MLI2 cells. ***P < 0.001, generalized linear mixed effect model (Supplementary Table 3). Right, representative traces of MLI1 (purple) and MLI2 (green) for 20, 40 and 60 pA current injections. **i**, *t*-SNE visualization of expression of *Gjd2* across MLI and PLI cell types. **j**, Representative image of smFISH expression of *Sorcs3* (purple), *Nxph1* (green), and *Gjd2* (red), in the molecular layer (*n* = 16 slides sectioned from 3 mice). Scale bar, 20 μm. **k**, The percentages of MLI1s and MLI2s expressing *Gjd2* in the molecular layer (total cells counted shown; *n* = 16 slides sectioned from 3 mice). Data are mean ± s.d. **l**, The percentages of all cells (grey), MLI1s (purple) and ML2s (green) in which spikelets were observed. **m**, Example voltage clamp recordings in the presence of synaptic blockers show spikelets in MLI1 (purple), but no spikelets in MLI2 (green). Firing rates, *R*m values, input–output curves and the numbers of cells with spikelets were all significantly different for MLI1 and MLI2 (Supplementary Table 3).

were more excitable than MLI1s (Fig. 4h), and displayed a stronger hyperpolarization-activated current (Extended Data Fig. 8).

MLIs are known to be electrically coupled via gap junctions[29], but it is not clear whether this is true for both MLI1s and MLI2s. In the cerebral cortex and some other brain regions, interneurons often electrically couple selectively to neurons of the same type, but not other types[30,31]. We therefore examined whether this also applies to MLI1s and MLI2s. The

expression of *Gjd2*, the gene encoding the dominant gap junction protein in MLIs[32], was found in MLI1s but not MLI2s, both in our single-nucleus data (Fig. 4i) and by smFISH (Fig. 4j, k), which suggests potential differences in electrical coupling. Notably, the two clusters of Golgi cells, another interneuron type known to be electrically coupled[33,34], differentially expressed many of the same markers, including *Sorcs3*, *Gjd2* and *Nxph1* in both human and mouse (Extended Data Figs. 4e, f, 9).

Action potentials in coupled MLIs produce small depolarizations known as spikelets that are thought to promote synchronous activity between MLIs[29]. We therefore investigated whether spikelets are present in MLI1s and absent in MLI2s. Consistent with the gene expression profile, we observed spikelets in 71% of MLI1s and not in MLI2s (Fig. 4l, m; $P < 0.001$, Fisher's exact test). These findings suggest that most MLI1s are coupled to other MLI1s by gap junctions, whereas MLI2s show no electrical coupling to other MLIs.

## Conclusions

In this Article, we used high-throughput, region-specific transcriptome sampling to build a comprehensive taxonomy of cell types in the mouse cerebellar cortex, and quantify spatial variation across individual regions. Our joint analyses with post mortem human samples indicated that the neuronal populations defined in mouse were generally conserved in human (Extended Data Fig. 4), consistent with a recent comparative analysis in the cerebral cortex[35]. We find considerably more regional specialization in PCs—especially in posterior lobules—than was previously recognized. These PC subtypes overlap with greater local abundances in UBCs and in distinct specializations in granule cells, which indicates a higher degree of regional circuit heterogeneity than previously thought. Our dataset is freely available to the neuroscience community (https://portal.nemoarchive.org/; https://singlecell.broadinstitute.org), facilitating functional characterization of these populations, many of which are entirely novel.

One of the biggest challenges facing the comprehensive cell typing of the brain is the correspondence problem[36]: how to integrate definitions of cell types on the basis of the many modalities of measurement used to characterize brain cells. We found success by first defining populations using systematic molecular profiling, and then relating these populations to physiological and morphological features using targeted, joint analyses of individual cells. We were surprised that the cerebellar MLIs—one of the first sets of neurons to be characterized more than 130 years ago[37]—are in fact composed of two molecularly and physiologically discrete populations, that each shows a similar morphological continuum along the depth axis of the molecular layer. As comprehensive cell typing proceeds across other brain regions, we expect the emergence of similar basic discoveries that challenge and extend our understanding of cellular specialization in the nervous system.

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

## Methods

### Animals

Nuclei suspensions for mouse (C57BL/6J, Jackson Labs) cerebellum profiles were generated from 2 female and 4 male adult mice (60 days old), 1 male E18 mouse, 1 male P0 (newborn) mouse, 1 female P4 (4 days old) mouse, 1 female P8, 2 male P12 and 2 female P16 mice. Adult mice were group-housed with a 12-h light-dark schedule and allowed to acclimate to their housing environment for two weeks after arrival. Timed pregnant mice were received and euthanized to yield E18 mice 6 days after arrival. Newborn mice were housed as individual litters for up to 16 days. All experiments were approved by and in accordance with Broad IACUC protocol number 012-09-16.

### Brain preparation

At E18, P0, P4, P8, P12, P16 and P60, C57BL/6J mice were anaesthetized by administration of isoflurane in a gas chamber flowing 3% isoflurane for 1 min. Anaesthesia was confirmed by checking for a negative tail and paw pinch response. Mice were moved to a dissection tray and anaesthesia was prolonged via a nose cone flowing 3% isoflurane for the duration of the procedure. Transcardial perfusions were performed on adult, pregnant (E18), P8, P12 and P16 mice with ice-cold pH 7.4 HEPES buffer containing 110 mM NaCl, 10 mM HEPES, 25 mM glucose, 75 mM sucrose, 7.5 mM MgCl$_2$, and 2.5 mM KCl to remove blood from the brain. P0 and P4 mice were unperfused. The brain was removed from P60, P8, P12 and P16 mice and frozen for 3 min in liquid nitrogen vapour. E18, P0 and P4 mice were sagittally bisected after similarly freezing their brains in situ. All tissue was moved to −80 °C for long-term storage. A detailed protocol is available at protocols.io (https://doi.org/10.17504/protocols.io.bcbrism6).

### Generation of cerebellar nuclei profiles

Frozen adult mouse brains were securely mounted by the frontal cortex onto cryostat chucks with OCT embedding compound such that the entire posterior half including the cerebellum and brainstem were left exposed and thermally unperturbed. Dissection of each of 16 cerebellar vermal and cortical lobules was performed by hand in the cryostat using an ophthalmic microscalpel (Feather safety Razor P-715) pre-cooled to −20 °C and donning four surgical loupes. Whole E18, P0, P4, P8, P12 and P16 mouse cerebella were similarly curated by dissecting rhombomeric cerebellar rudiments from sagittal frozen brain hemispheres using a pre-cooled 1-mm disposable biopsy punch (Integra Miltex). Each excised tissue dissectate was placed into a pre-cooled 0.25 ml PCR tube using pre-cooled forceps and stored at −80 °C. Nuclei were extracted from this frozen tissue using gentle, detergent-based dissociation, according to a protocol available at protocols.io (https://doi.org/10.17504/protocols.io.bck6iuze) adapted from one provided by the McCarroll laboratory (Harvard Medical School), and loaded into the 10x Chromium V3 system. Reverse transcription and library generation were performed according to the manufacturer's protocol.

### Floating slice hybridization chain reaction on acute slices

Acute cerebellar slices containing Alexa 594-filled patched cells were fixed as described and stored in 70% ethanol at 4 °C until hybridization chain reaction (HCR). They were then subjected to a 'floating slice HCR' protocol in which the recorded cells could be simultaneously re-imaged in conjunction with HCR expression analysis in situ and catalogued as to their positions in the cerebellum. A detailed protocol (https://doi.org/10.17504/protocols.io.bck7iuzn) was performed using the following HCR probes and matching hairpins purchased from Molecular Instruments: glutamate metabotropic receptor 8 (*Grm8*) lot number PRC005, connexin 36 (*Gjd2*) lot number PRD854 and PRA673, cadherin22 (*Cdh22*) lot number PRC011, neurexophilin 1 (*Nxph1*) lot number PRC675 and PRC466, leucine-rich glioma-inactivated protein 2 (*Lgi2*) lot number PRC012, somatostatin (*Sst*) lot number PRA213

and sortilin related VPS10 domain containing receptor 3 (*Sorcs3*) lot number PRC004. Amplification hairpins used were type B1, B2 and B3 in 488 nm, 647 nm and 546 nm respectively.

### Patch fill and HCR co-imaging

After floating-slice HCR, slices were mounted between no.1 coverslips with antifade compound (ProLong Glass, Invitrogen) and images were collected on an Andor CSU-X spinning disk confocal system coupled to a Nikon Eclipse Ti microscope equipped with an Andor iKon-M camera. The images were acquired with an oil immersion objective at 60×. The Alexa 594 patched cell backfill channel (561 nm) plus associated HCR probe/hairpin channels (488 nm and 647 nm) were projected through a 10–20-µm thick z-series so that an unambiguous determination of the association between the patch-filled cell and its HCR gene expression could be made. Images were processed using Nikon NIS Elements 4.4 and Nikon NIS AR.

### Human brain and nuclei processing

Human donor tissue was supplied by the Human Brain and Spinal Fluid Resource Center at UCLA, through the NIH NeuroBioBank. This work was determined by the Office of Research Subjects Protection at the Broad Institute not to meet the definition of human subjects research (project ID NHSR-4235).

Nuclei suspensions from human cerebellum were generated from two neuropathologically normal control cases—one female tissue donor, aged 35, and one male tissue donor, aged 36. These fresh frozen tissues had post mortem intervals of 12 and 13.5 h respectively, and were provided as whole cerebella cut into four coronal slabs. A sub-dissection of frozen cerebellar lobules was performed on dry ice just before 10x processing and nuclei were extracted from this frozen tissue using gentle, detergent-based dissociation, according to a protocol available at protocols.io (https://doi.org/10.17504/protocols.io.bck6iuze).

### Electrophysiology experiments

Acute parasagittal slices were prepared at 240-µm thickness from wild-type mice aged P30–P50. Mice were anaesthetized with an intraperitoneal injection of ketamine (10 mg kg$^{-1}$), perfused transcardially with an ice-cold solution containing (in mM): 110 choline chloride, 7 MgCl$_2$, 2.5 KCl, 1.25 NaH$_2$PO$_4$, 0.5 CaCl$_2$, 25 glucose, 11.5 sodium ascorbate, 3 sodium pyruvate, 25 NaHCO$_3$, 0.003 (*R*)-CPP, equilibrated with 95% O$_2$ and 5% CO$_2$. Slices were cut in the same solution and were then transferred to artificial cerebrospinal fluid (ACSF) containing (in mM) 125 NaCl, 26 NaHCO$_3$, 1.25 NaH$_2$PO$_4$, 2.5 KCl, 1 MgCl$_2$, 1.5 CaCl$_2$ and 25 glucose equilibrated with 95% O$_2$ and 5% CO$_2$ at approximately 34 °C for 30 min. Slices were then kept at room temperature until recording.

All UBC recordings were done at 34–36 °C with (in µM) 2 (*R*)-CPP, 5 NBQX, 1 strychnine, 10 SR95531 (gabazine) and 1.5 CGP in the bath to isolate metabotropic currents. Loose cell-attached recordings were made with ACSF-filled patch pipettes of 3–5 MΩ resistance. Whole-cell voltage-clamp recordings were performed while holding the cell at −70 mV with an internal containing (in mM): 140 KCl, 4 NaCl, 0.5 CaCl$_2$, 10 HEPES, 4 MgATP, 0.3 NaGTP, 5 EGTA 5, and 2 QX-314, pH adjusted to 7.2 with KOH. Brief puffs of glutamate (1 mM for 50 ms at 5 psi) were delivered using a Picospritzer II (General Valve Corp.) in both cell-attached and whole-cell configuration to assure consistent responses. The heat map of current traces from all cells are sorted by the score over the first principal axis after singular value decomposition (SVD) of recordings over all cells.

For whole-cell recordings with pharmacology, we used an K-methanesulfonate internal containing (in mM): 122 K-methanesulfonate, 9 NaCl, 9 HEPES, 0.036 CaCl$_2$, 1.62 MgCl$_2$, 4 MgATP, 0.3 GTP (Tris salt), 14 creatine phosphate (Tris salt), and 0.18 EGTA, pH 7.4. A junction potential of −8 mV was compensated for during recording. 300 nM TTX was added to the ACSF in conjunction with the synaptic blockers listed above. Three pipettes filled with ACSF

containing 1 mM glutamate, 100 µM DHPG or 1 µM LY354740 were positioned within 20 µm of the recorded cell. Pressure applications of each agonist were delivered at 10 psi with durations of 40–50 ms. Agonist applications were separated by 30 s. Two to three trials were collected for each agonist.

MLI recordings were performed at approximately 32 °C with an internal solution containing (in mM) 150 K-gluconate, 3 KCl, 10 HEPES, 3 MgATP, 0.5 GTP, 5 phosphocreatine-tris$_2$ and 5 phosphocreatine-Na$_2$, 2 mg ml$^{-1}$ biocytin and 0.1 Alexa 594 (pH adjusted to 7.2 with KOH, osmolality adjusted to 310 mOsm kg$^{-1}$). Visually guided whole-cell recordings were obtained with patch pipettes of around 4 MΩ resistance pulled from borosilicate capillary glass (BF150-86-10, Sutter Instrument). Electrophysiology data was acquired using a Multiclamp 700B amplifier (Axon Instruments), digitized at 20 kHz and filtered at 4 kHz. For isolating spikelets in MLI recordings, cells were held at −65 mV in voltage clamp and the following receptor antagonists were added to the solution (in µM) to block synaptic currents: 2 (R)-CPP, 5 NBQX, 1 strychnine, 10 SR95531 (gabazine), 1.5 CGP. All drugs were purchased from Abcam and Tocris. To obtain an input-output curve, MLIs were maintained at 60–65 mV with a constant hyperpolarizing current, and 250 ms current steps ranging from −30 pA to +100 pA were injected in 10 pA increments. To activate the hyperpolarization-evoked current ($I_h$), MLIs were held at −65 mV and a 30 pA hyperpolarizing current step of 500 ms duration was injected. The amplitude of $I_h$ was calculated as the difference between the maximal current evoked by the hyperpolarizing current step and the average steady-state current at the end (480–500 ms) of the current step. Capacitance and input resistance ($R_i$) were determined using a 10 pA, 50 ms hyperpolarizing current step. To prevent excessive dialysis and to ensure successful detection of mRNAs in the recorded cells, the total duration of recordings did not exceed 10 min. Acquisition and analysis of electrophysiological data were performed using custom routines written in MATLAB (Mathworks), IgorPro (Wavemetrics), or AxoGraphX. Data are reported as median ± interquartile range, and statistical analysis was carried out using the Mann–Whitney or Fisher's exact test, as indicated. Statistical significance was assumed at $P < 0.05$.

To determine the presence of spikelets, peak detection was used to generate event-triggered average waveforms with thresholds based on the mean absolute deviation (MAD) of the raw trace. Spikelet recordings were scored for the presence of spikelets blind to the molecular identity of the cells. The analysis was restricted to cells recorded in the presence of synaptic blockers.

## Imaging and analysis

MLIs were filled with 100 µM Alexa-594 via patch pipette to visualize their morphology using two-photon imaging. After completion of the electrophysiological recordings the patch electrode was retracted slowly and the cell resealed. We used a custom-built two-photon laser-scanning microscope with a 40×, 0.8 numerical aperture (NA) objective (Olympus Optical) and a pulsed two-photon laser (Chameleon or MIRA 900, Coherent, 800 nm excitation). DIC images were acquired at the end of each experiment and locations of each cell within the slice were recorded. Two-photon images were further processed in ImageJ.

## Tissue fixation of acute slices

After recording and imaging, cerebellar slices were transferred to a well-plate and submerged in 2–4% PFA in PBS (pH 7.4) and incubated overnight at 4 °C. Slices were then washed in PBS (3 × 5 min) and then kept in 70% ethanol in RNase-free water until HCR was performed.

## Preprocessing of sequencing reads

Sequencing reads from mouse cerebellum experiments were demultiplexed and aligned to a mouse (mm10) premrna reference using CellRanger v3.0.2 with default settings. Digital gene expression matrices were generated with the CellRanger count function. Sequencing reads from human cerebellum experiments were demultiplexed and aligned to a human (hg19) premrna reference using the Drop-seq alignment workflow[2], which was also used to generate the downstream digital gene expression matrices.

## Estimation of adequate rare cell type detection

To estimate the probability of sufficiently sampling rare cell types in the cerebellum as a function of total number of nuclei sampled, we used the approach proposed by the Satija laboratory (https://satijalab.org/howmanycells), with the assumption of at most 10 very rare cell types, each with a prevalence of 0.15%. We derived this minimum based on the observed prevalences of the two rarest cell types we identified (OPC_4, Purkinje_Aldoc_2). We set 70 cells as the threshold for sufficient sampling, and calculated the overall probability as a negative binomial (NB) density:

$$NB(k; n, p)^m$$

in which $k = 70$, $P = 0.0015$, $m = 10$, and $n$ represents the total number of cells sampled.

## Cell type clustering and annotation

After generation of digital gene expression matrices as described above, we filtered out nuclei with fewer than 500 UMIs. We then performed cell type annotation iteratively through a number of rounds of dimensionality reduction, clustering, and removal of putative doublets and cells with high mitochondrial expression. For the preliminary clustering step, we performed standard preprocessing (UMI normalization, highly variable gene selection, scaling) with Seurat v2.3.4 as previously described[38]. We used principal component analysis (PCA) with 30 components and Louvain community detection with resolution 0.1 to identify major clusters (resulting in 34 clusters). At this stage, we merged several clusters (primarily granule cell clusters) based on shared expression of canonical cell type markers, and removed one cluster whose top differentially expressed genes were mitochondrial (resulting in 11 clusters).

For subsequent rounds of cluster annotation within these major cell type clusters, we applied a variation of the LIGER workflow previously described[12], using integrative non-negative matrix factorization (iNMF) to limit the effects of sample- and sex-specific gene expression. In brief, we normalized each cell by the number of UMIs, selected highly variable genes[7] and spatially variable genes (see section below), performed iNMF, and clustered using Louvain community detection (omitting the quantile normalization step). Clusters whose top differentially expressed genes indicated contamination from a different cell type or high expression of mitochondrial genes were removed during the annotation process, and not included in subsequent rounds of annotation. This iterative annotation process was repeated until no contaminating clusters were identified in a round of clustering. Differential expression analysis within rounds of annotation was performed with the Wilcoxon rank sum test using Seurat's FindAllMarkers function. Comprehensive differential expression analysis across all 46 final annotated clusters was performed using the wilcoxauc function from the presto package[39]. A full set of parameters used in the LIGER annotation steps and further details can be found in Supplementary Table 4.

For visualization as in Fig. 1b, we merged all annotated high-quality nuclei and repeated preliminary preprocessing steps before performing UMAP using 25 principal components.

## Integrated analysis of human and mouse data

After generation of digital gene expression matrices for the human nuclei profiles, we filtered out nuclei with fewer than 500 UMIs. We then performed a preliminary round of cell type annotation using the standard LIGER workflow (integrating across batches) to identify the primary human interneuron populations (UBCs, MLIs and PLIs, Golgi cells,

granule cells; based on the same markers as in Supplementary Table 1). We repeated an iteration of the same workflow for the four cell populations specified above (with an additional quantile normalization step) in order to identify and remove putative doublet and artefactual populations. Finally, we performed iNMF metagene projection as previously described[40] to project the human datasets into latent spaces derived from the corresponding mouse cell type datasets. We then performed quantile normalization and Louvain clustering, assigning joint clusters based on the previously annotated mouse data clusters. For the granule cell joint analysis, we first limited the mouse data to include only the five cerebellar regions sampled in human data collection (lobules II, VII, VIII, IX and X). For the Golgi cell joint analysis, we performed iNMF (integrating across species), instead of metagene projection.

### Spatially variable gene selection

To identify genes with high regional variance, we first computed the log of the index of dispersion (log variance-to-mean ratio, logVMR) for each gene, across each of the 16 lobular regions. Next, we simulated a Gaussian null distribution whose centre was the logVMR mode, found by performing a kernel density estimation of the logVMRs (using the density function in R, followed by the turnpoints function). The standard deviation of the Gaussian was computed by reflecting the values less than the mode across the centre. Genes whose logVMRs were in the upper tail with $P < 0.01$ (Benjamini–Hochberg adjusted) were ruled as spatially variable. For the granule cell and PC cluster analyses, adjusted $P$-value thresholds were set to 0.001 and 0.002, respectively.

### Cluster regional composition testing and lobule enrichment

To determine whether the lobule composition of a cluster differs significantly from the corresponding outer level cell type lobule distribution, we used a multinomial test approximated by Pearson's chi-squared test with $k − 1$ degrees of freedom, in which $k$ was the total number of lobules sampled (16). The expected number of nuclei for a cluster $i$ and lobule $j$ was estimated as follows:

$$E_{ij} = N_i \times \frac{N_j}{\sum_j N_j}$$

where $N_i$ is the total number of nuclei in cluster $i$ and $N_j$ is the number of nuclei in lobule $j$ (across all clusters in the outer level cell type, as defined below). The resulting $P$ values were FDR-adjusted (Benjamini–Hochberg) using the p.adjust function in R.

Lobule enrichment (LE) scores for each cluster $i$ and each lobule $j$ were calculated by:

$$LE_{ij} = \frac{\frac{n_{ij}}{\sum_j n_{ij}}}{\frac{N_j}{\sum_j N_j}}$$

in which $n_{ij}$ is the observed number of nuclei in cluster $i$ and lobule $j$, and $N_j$ is the number of nuclei in lobule $j$ (across all clusters in the outer level cell type). For this analysis, we used coarse cell type definitions shown coloured in the Fig. 2a, and merged the PLI clusters. For lobule composition testing and replicate consistency analysis below, we downsampled granule cells to 60,000 nuclei (the next most numerous cell type were the MLI and PLI clusters with 45,767 nuclei).

To determine the consistency of lobule enrichment scores across replicates in each region, we designated two sets of replicates by assigning nuclei from the most represented replicate in each region and cluster analysis to 'replicate 1' and nuclei from the second most represented replicate in each region to 'replicate 2'. This assignment was used because not all regions had representation from all individuals profiled, and some had representation from only two individuals. We calculated lobule enrichment scores for each cluster using each of the replicate sets separately; we then calculated the Pearson correlation between

the two sets of lobule enrichment scores for each cluster. We would expect correlation to be high for clusters when lobule enrichment is biologically consistent. We note that one cluster (Purkinje_Aldoc_2), was excluded from the replicate consistency analysis as under this design, it had representation from only a single aggregated replicate. However, we confirmed that lobule enrichment for this cluster was strongly consistent with Allen Brain Atlas expression staining (Extended Data Fig. 3c).

### Continuity of gene expression

To characterize molecular variation across cell types, we attempted to quantify the continuity of scaled gene expression across a given cell type pair, ordered by pseudotime rank (calculated using Monocle2). For each gene, we fit a logistic curve to the scaled gene expression values and calculated the maximum slope ($m$) of the resulting curve, after normalizing for both the number of cells and dynamic range of the logistic fit. To limit computational complexity, we downsampled cell type pairs to 5,000 total nuclei.

We fit curves and computed $m$ values for the most significantly differentially expressed genes across five cell type pairs (Fig. 3b). Differentially expressed genes were determined using Seurat's FindMarkers function. We then plotted the cumulative distribution of $m$ values for the top 200 genes for each cell type pair; genes were selected based on ordering by absolute Spearman correlation between scaled gene expression and pseudotime rank.

### Trajectory analysis of peri- and postnatal mouse cerebellum data

After generation of digital gene expression matrices for the peri- and postnatal mouse profiles, we filtered out nuclei with fewer than 500 UMIs. We applied the LIGER workflow (similarly to the adult mouse data analysis), to identify clusters corresponding to major developmental pathways. We then isolated the cluster corresponding to GABAergic progenitors (marked by expression of *Tfap2b* and other canonical markers). We performed a second iteration of LIGER iNMF and Louvain clustering on this population and generated a UMAP representation. Using this UMAP representation, we calculated pseudotime ordering and a corresponding trajectory graph with Monocle3[41]. To identify modules of genes which varied along the computed trajectory, we used the graph_test and find_gene_modules functions from Monocle3.

### Reporting summary

Further information on research design is available in the Nature Research Reporting Summary linked to this paper.

## Data availability

All processed data and annotations have been made freely available for download and visualization through an interactive Single Cell Portal study (https://singlecell.broadinstitute.org/single_cell/study/SCP795/). Raw and processed data that support the findings of this study have been deposited in Gene Expression Omnibus (GEO) under accession number GSE165371 and in the Neuroscience Multi-omics (NeMO) Archive (https://nemoarchive.org/).

## Code availability

Code and scripts to reproduce analyses presented here are available on Github at https://github.com/MacoskoLab/cerebellum-atlas-analysis.

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

**Acknowledgements** We thank K. Allaway, G. Fishell and A. Goeva for discussions and advice. We thank J. Langlieb for his assistance with data preparation and analysis. This work was supported by the Stanley Center for Psychiatric Research, NIH/NIMH Brain Grant 1U19MH114821 to E.Z.M., an NIH New Innovator Award to E.Z.M. (DP2AG058488), and by NIH/NINDS R35NS097284 to W.G.R.

**Author contributions** V.K. performed analyses, with help from C.M. C.M. generated the single-nucleus profiles, with help from N.N. C.G. and W.R. conceived UBC physiology experiments, which were performed by C.G. W.R., T.O. and S.R. conceived MLI electrophysiology and imaging experiments, which were performed and analysed by T.O. and S.R. C.V. performed dissections. N.N., C.V. and C.M. performed the smFISH experiments. E.Z.M. conceived the molecular study, with help from A.R. E.Z.M. and V.K. wrote the paper with contributions from all authors.

**Competing interests** A.R. is a founder and equity holder of Celsius Therapeutics, an equity holder in Immunitas Therapeutics and until 31 August 2020, was an SAB member of Syros Pharmaceuticals, Neogene Therapeutics, Asimov and ThermoFisher Scientific. From 1 August 2020, A.R. has been an employee of Genentech. For E.Z.M., the research herein was not done in his capacity as an MGH employee.

**Additional information**
**Correspondence and requests for materials** should be addressed to E.M.

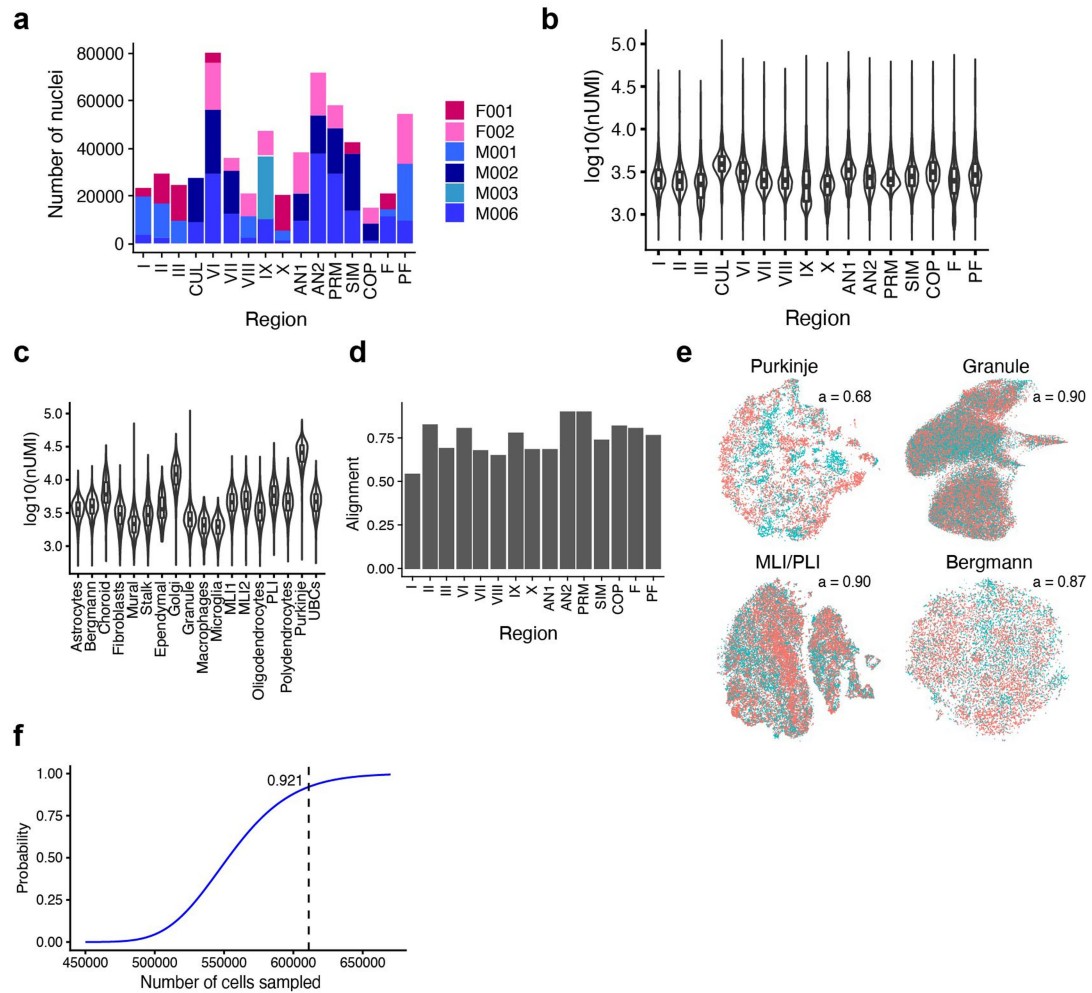

**Extended Data Fig. 1 | Summary and quality control analyses for nuclei sampling. a**, Bar graph showing the number of cells contributed by each individual per region across the dataset of 611,034 cerebellar nuclei (post-quality control, 6 total individuals, 16 regions). **b**, Violin plot showing distribution of log₁₀(nUMI) across the regions profiled. Cell numbers: $n = 23,364$ for region I, $n = 29,513$ for region II, $n = 24,344$ for region III, $n = 27,680$ for region CUL, $n = 80,166$ for region VI, $n = 35,622$ for region VII, $n = 21,299$ for region VIII, $n = 47,118$ for region IX, $n = 20,423$ for region X, $n = 38,227$ for region AN1, $n = 72,135$ for region AN2, $n = 58,142$ for region PRM, $n = 42,518$ for region SIM, $n = 14,816$ for region COP, $n = 20,836$ for region F, $n = 54,831$ for region PF. Box plots centred on median and bounded by interquartile range. **c**, Violin plot of log₁₀(nUMI) per profile across the 18 cell types identified. The relative median values here are consistent with known differences in cell size; for example, Purkinje cells have the highest median number of UMIs. Cell numbers: $n = 16,717$ for astrocytes, $n = 17,498$ for Bergmann glia, $n = 591$ for choroid, $n = 5,333$ for fibroblasts, $n = 2,271$ for mural cells, $n = 6,142$ for stalk cells, $n = 243$ for ependymal cells, $n = 3,989$ for Golgi cells, $n = 477,176$ for granule cells, $n = 280$ for macrophages, $n = 1,296$ for

microglia, $n = 32,716$ for MLI1, $n = 10,608$ for MLI2, $n = 13,363$ for oligodendrocytes, $n = 2,443$ for PLIs, $n = 2,121$ for polydendrocytes, $n = 16,634$ for Purkinje cells, $n = 1,613$ for UBCs. Box plots centred on median and bounded by interquartile range. **d**, Bar graph of alignment scores (Methods) calculated across replicates for each lobule, after performing LIGER integration (across sex) (Methods) for each regional subset. Subsets sampled from the final set of 611,034 high-quality nuclei profiles. These analyses represent examples of expected replicate alignment when using the described pipeline. Note that lobule COP is excluded as it did not include representation from male and female replicates. **e**, Visualizations of representative cell type analyses, indicating high alignment across replicate sets (granule is UMAP, all others *t*-SNE). Replicate sets were designated as in Methods (cluster regional composition test and lobule enrichment). **f**, Plot indicating probability of sufficiently sampling 10 very rare populations (prevalence 0.15%) as a function of total number of cells profiled in experiment (probability estimated as in Methods). Number of high-quality nuclei profiled here (611,034) and corresponding probability are indicated.

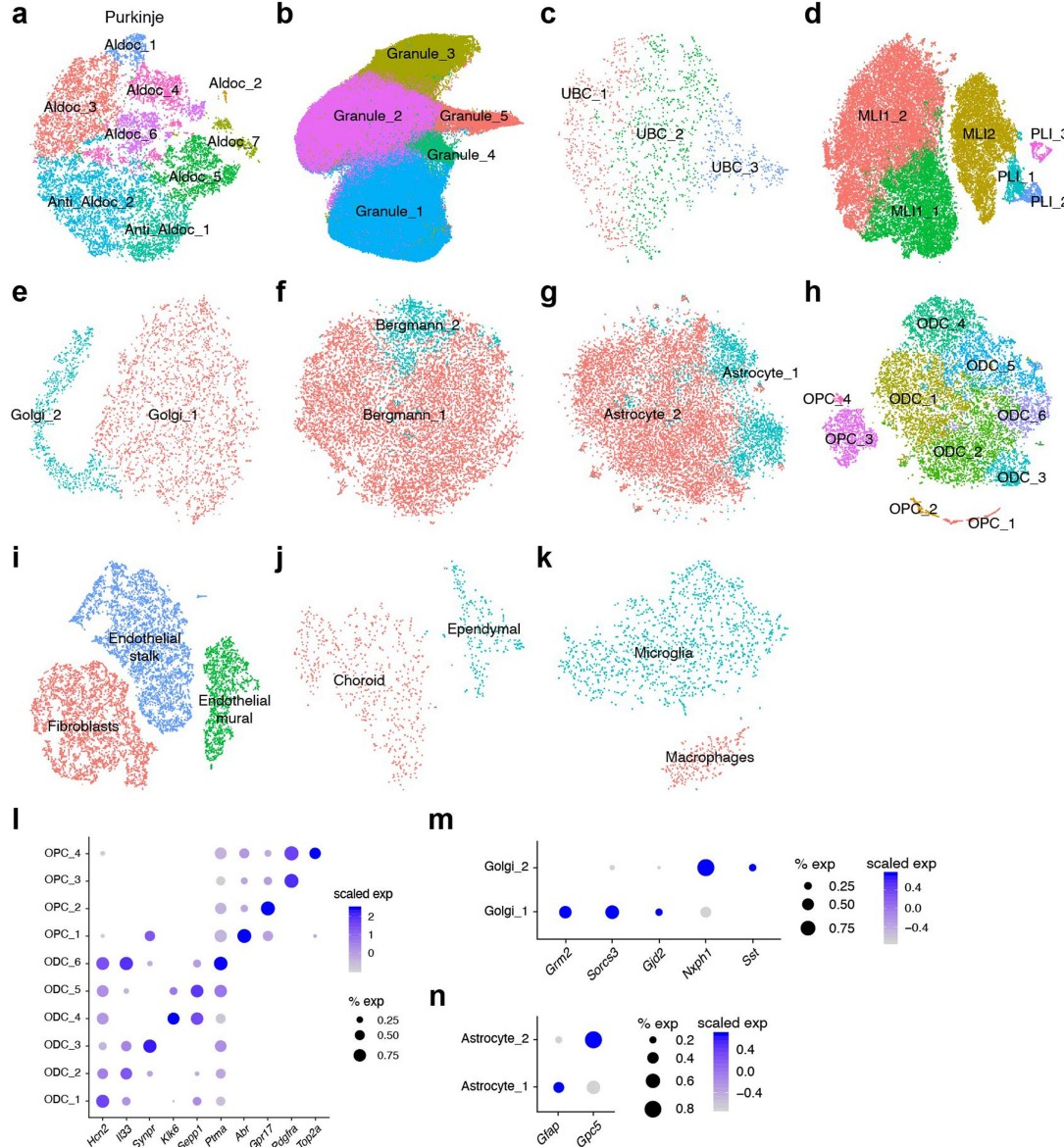

**Extended Data Fig. 2 | Characterization and annotation of cerebellar subtypes. a–k**, Visualizations of all individual cell type analyses of Purkinje (**a**), granule (**b**), UBC (**c**), MLI or PLI (**d**), and Golgi (**e**) neurons, as well as Bergmann (**f**), astrocyte (**g**), OPC/ODC (**h**), endothelial (**i**), choroid (**j**), and macrocytic (**k**) glial populations, labelled by cluster designations. Granule is UMAP, all others *t*-SNE. **l–n**, Dot plots of scaled expression of selected marker genes for individual cell type analyses not displayed in the main figures: ODC and OPC (**l**), Golgi (**m**) and astrocyte (**n**).

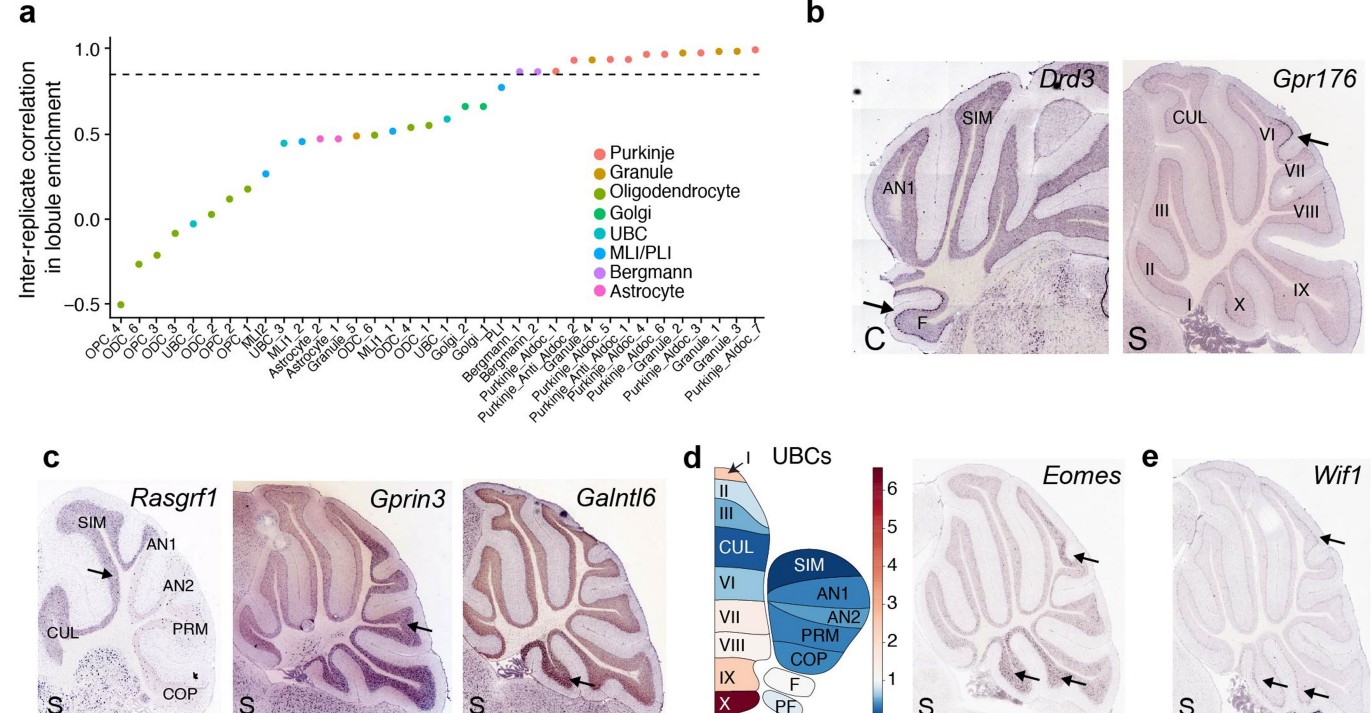

**Extended Data Fig. 3 | Additional analyses of spatial variation in neuronal and glial subtypes. a**, Scatter plot of inter-replicate correlation (Pearson) for lobule enrichment scores calculated for replicate sets individually, across each cluster (clusters ordered by decreasing correlation). Two replicate sets were designated for each major cluster analysis by aggregating the individuals with the highest representation for each lobule into a single replicate (and similarly for the individuals with second highest representation). High inter-replicate correlation indicates consistent lobule enrichment for subtypes. $r = 0.85$ is indicated. **b**, Allen Brain Atlas expression staining for two selected PC markers representing clusters with their respective lobule enrichments indicated; *Drd3* in the flocculus (Purkinje_Aldoc_2), and *Gpr176* in lobule VI (Purkinje_Aldoc_1).

C, coronal section; S, sagittal section. **c**, Allen Brain Atlas expression staining for three selected GC markers representing clusters with their respective lobule enrichments indicated; *Rasgrf1* in the anterior lobules (Granule_1), *Gprin3* in the posterior lobules (Granule_2), and *Galntl6* in the nodulus (Granule_3). **d**, Left, lobule enrichment plot indicating enrichment of UBCs in posterior lobules of the cerebellum, particularly lobules IX (uvula) and X (nodulus). Right, Allen Brain Atlas expression staining for UBC marker *Eomes*, showing enrichment in lobules IX, X and VII. **e**, Allen Brain Atlas expression staining for *Wif1* (a Bergmann_2 cluster marker), indicating expression enriched in lobules VI, X (nodulus), and IX (uvula).

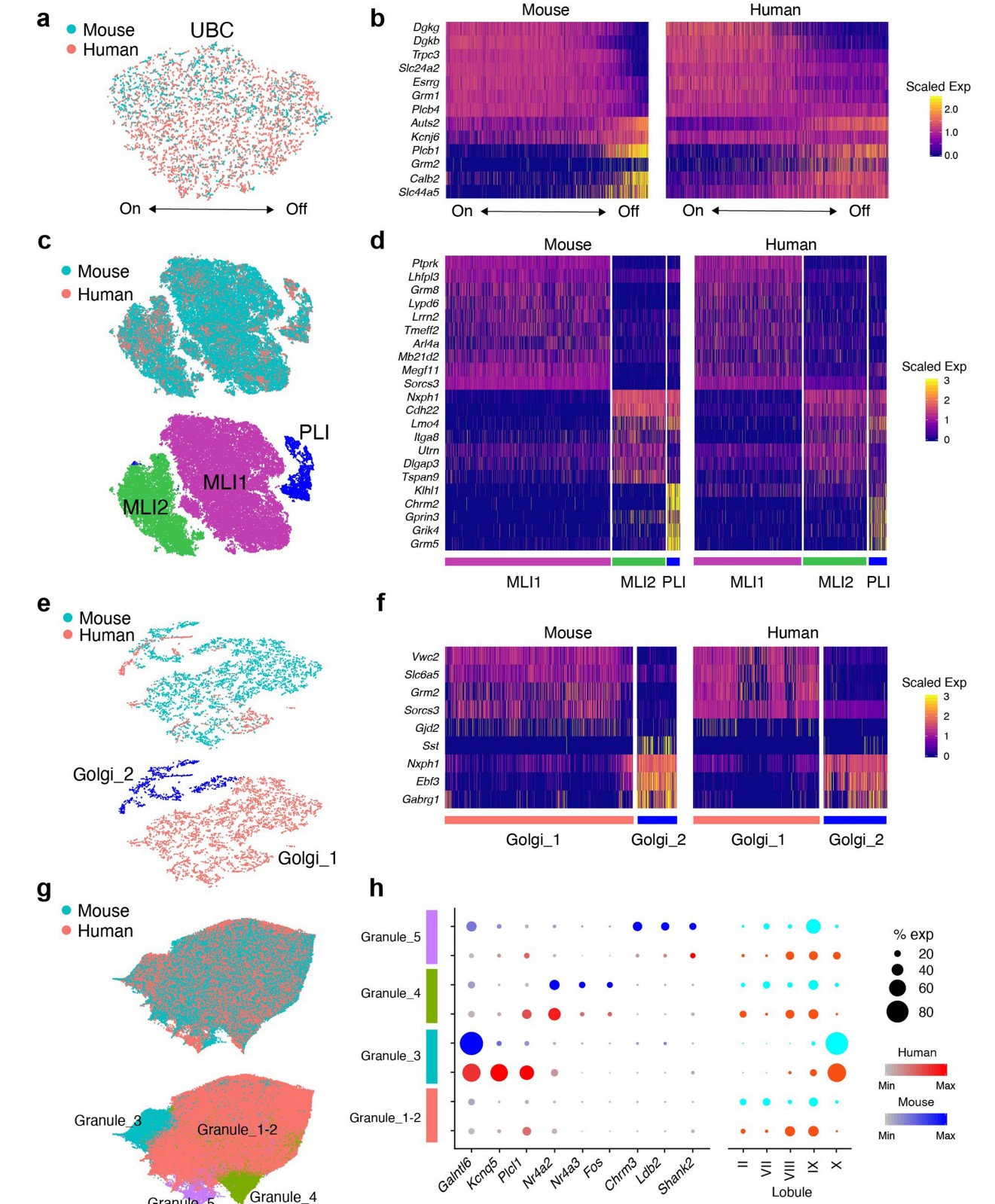

**Extended Data Fig. 4 | Integrative analysis of human and mouse cerebellar interneuron profiles. a**, **c**, **e**, **g**, UMAP representation of the integrative analyses of UBC (1,613 mouse; 3,893 human) (**a**), MLI/PLI (45,555 mouse; 14,971 human) (**c**), Golgi (3,989 mouse; 1,059 human) (**e**), and granule (119,972 mouse; 130,335 human) (**g**) cells, coloured by species (top), or joint cluster (bottom, for MLI and PLI, Golgi, and granule only). **b**, **d**, **f**, Heat maps showing expression of selected genes in UBC (**b**), MLI and PLI (**d**), and Golgi (**f**). Profiles are segregated both by species and cluster. UBC profiles are ordered by iNMF factor loadings for factor corresponding to OFF UBCs. **h**, Left, dot plot showing expression of selected genes in granule clusters, within human (red) and mouse (blue). Right, proportional representation of lobule dissections across the granule clusters. Granule cluster numbers approximately correspond to the mouse-only clusters shown in Fig. 2e.

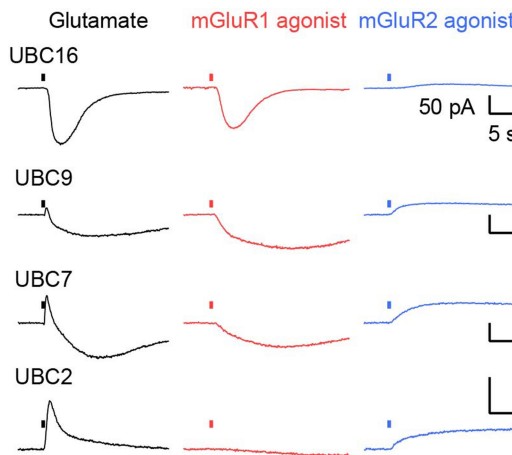

**Extended Data Fig. 5 | Recordings of UBCs.** Representative recordings of UBCs from Fig. 3f. In cells in which glutamate evoked primarily an inward current (UBC16), there was a very large mGluR1 component and a small mGluR2 component. The opposite was true for UBCs in which glutamate evoked primarily an outward current (UBC2). For intermediate cells such as UBC9 and UBC7, mGluR1 and mGluR2 components were both prominent.

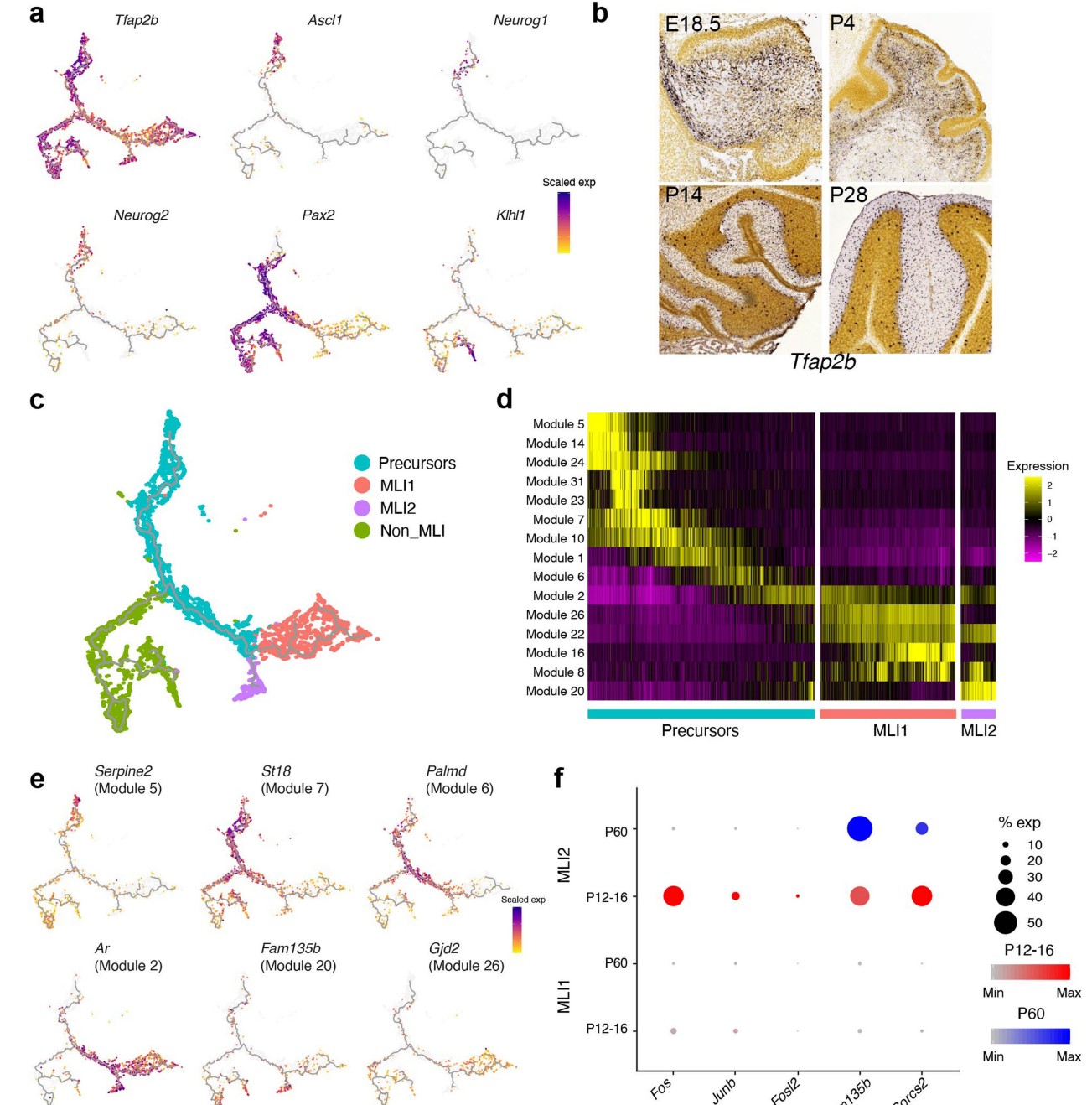

**Extended Data Fig. 6 | Developmental trajectory analysis of MLI1 and MLI2 neurons. a**, UMAP visualizations showing expression of canonical marker genes of cerebellar interneuron development (*Tfap2b*, *Ascl1*, *Neurog1*, *Neurog2* and *Pax2*), and of mature PLIs (*Klhl1*). **b**, Allen Developing Mouse Brain Atlas expression staining of *Tfap2b* across four developmental time points, showing that MLI progenitors begin to enter the molecular layer around P14. **c**, Interneuron developmental trajectory coloured by annotated clusters.

**d**, Heat map showing the loading of Monocle3-defined gene modules across undifferentiated interneuron precursors, MLI1 and MLI2. Cells are ordered in each block by pseudotime rank. **e**, UMAP visualizations showing expression of selected top-loading genes from the indicated Monocle3-defined modules. **f**, Dot plot showing the transient, MLI2-specific expression of three activity-regulated genes (*Fos*, *Junb* and *Fosl2*) and two genes whose expression persists in adulthood (*Fam135b* and *Sorcs2*).

**a**

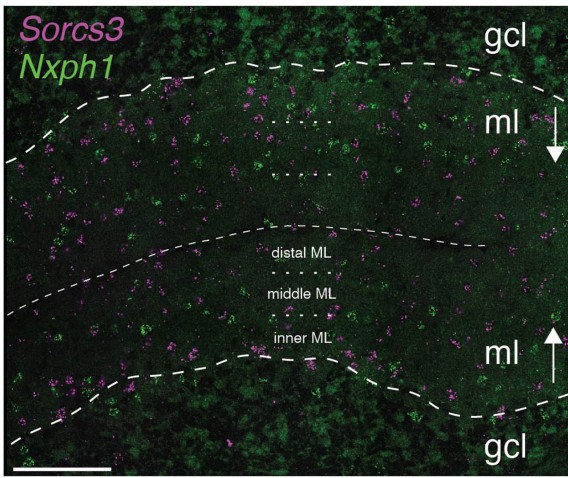

**b**

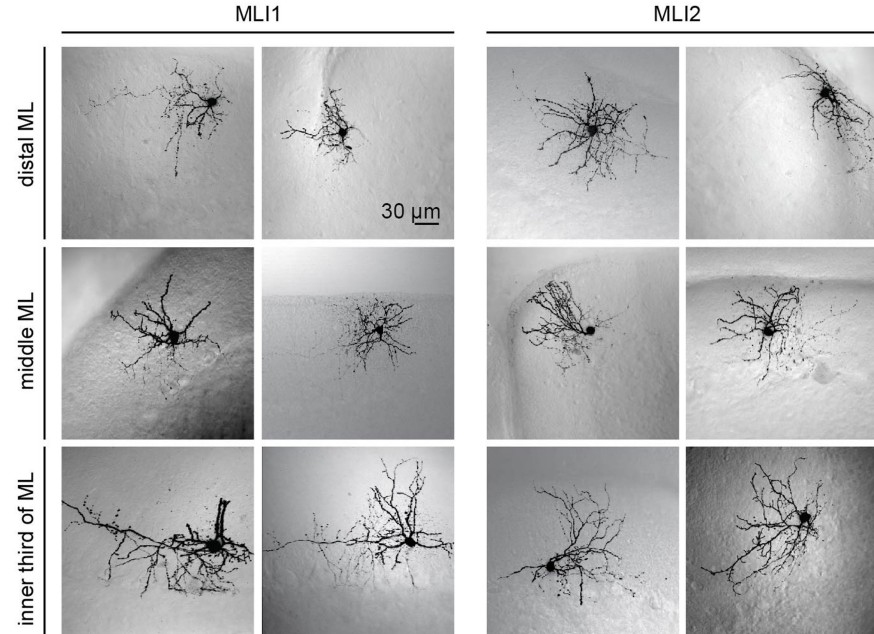

**Extended Data Fig. 7 | Additional imaging of MLI1 and MLI2 neurons.**
**a**, Representative image of smFISH expression of *Sorcs3* (purple) and *Nxph1* (green) at low magnification. Scale bar, 100 μm. The distal, middle and inner molecular layer sublayer boundaries are designated with a dotted line; the thinner dashed line marks the outer boundary of cortical folds, and the thicker dashed line indicates the location of the Purkinje layer (*n* = 16 slides sectioned over 3 mice). **b**, MLIs were imaged and identified as in Fig. 4. Examples of MLI1s (left) and MLI2s (right) are shown for cells located in the distal third (top), the middle third (middle) and the inner third (bottom) of the molecular layer. *n* = 23 MLI1 cells and 20 MLI2 cells.

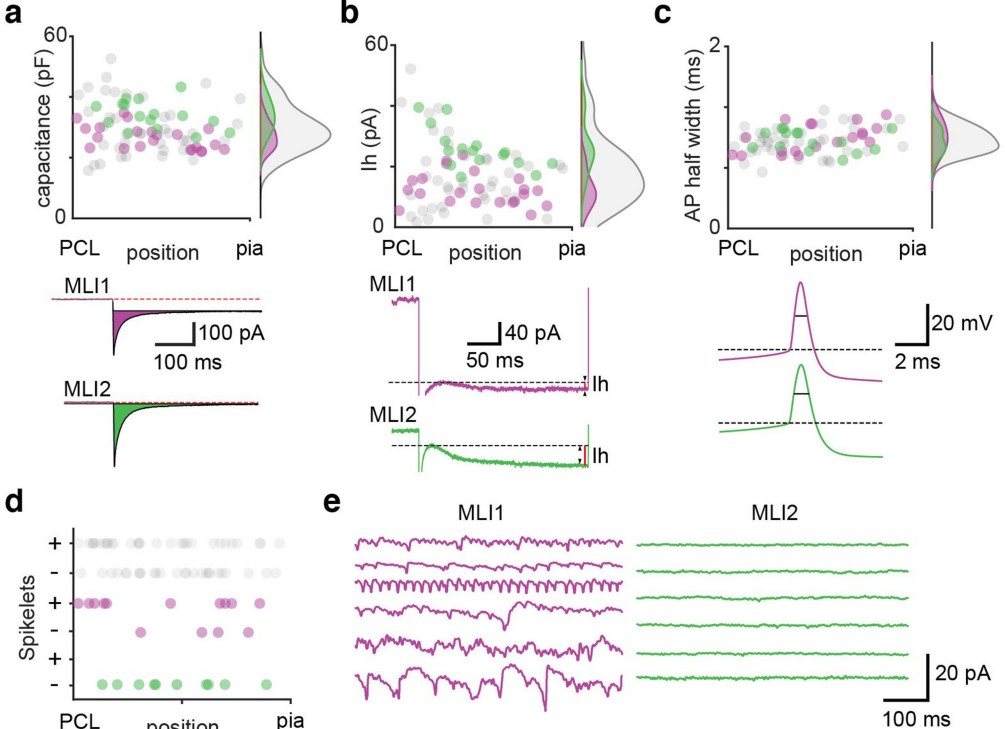

**Extended Data Fig. 8 | Further comparison of electrical properties of MLI1 and MLI2 neurons. a–c,** Measurements of: capacitance (**a**), the amplitude of currents through hyperpolarization and nucleotide-gated (HCN) channels, $I_h$, (**b**) and the action potential width (**c**) in recordings of subsequently identified MLI1 (purple) and MLI2 (green) neurons. The properties are plotted as a function of position within the molecular layer. Density plots summarize the properties for all cells (grey), MLI1 (purple) and MLI2 (green). **a,** Capacitance was determined by measuring the responses to a −10-mV voltage step, and integrating the capacitive current as shown (purple and green shaded area for MLI1 and MLI2, respectively). These traces from cells in the inner third of the molecular layer show the large difference in resistance ($R_m = −10$ mV $I_{ss}$), in which $I_{ss}$ is the steady state current in response to the voltage step (black continuous line). Red dashed line denotes the baseline current. There was a significant difference in the capacitance of MLI1 and MLI2 neurons ($P = 2.03 \times 10^{-4}$). **b,** The amplitude of $I_h$ was determined by measuring the responses to a −30-mV step, and evaluated as shown. Measured $I_h$ was significantly larger for MLI2 ($P = 4.24 \times 10^{-6}$), and the difference was particularly notable for MLIs in the inner third of the molecular layer. **c,** The action potential width was measured as shown and there was no significant difference for MLI1 and MLI2 neurons. **d,** The presence or absence of spikelets is shown as a function of position in the molecular layer is summarized for all MLIs, MLI1 and MLI2, with '+' indicating the presence of spikelets. **e,** Example recordings are shown for six MLI1 neurons (left, purple) and six MLI2 neurons (right, green). Number of cells and statistical tests are summarized in Supplementary Table 3.

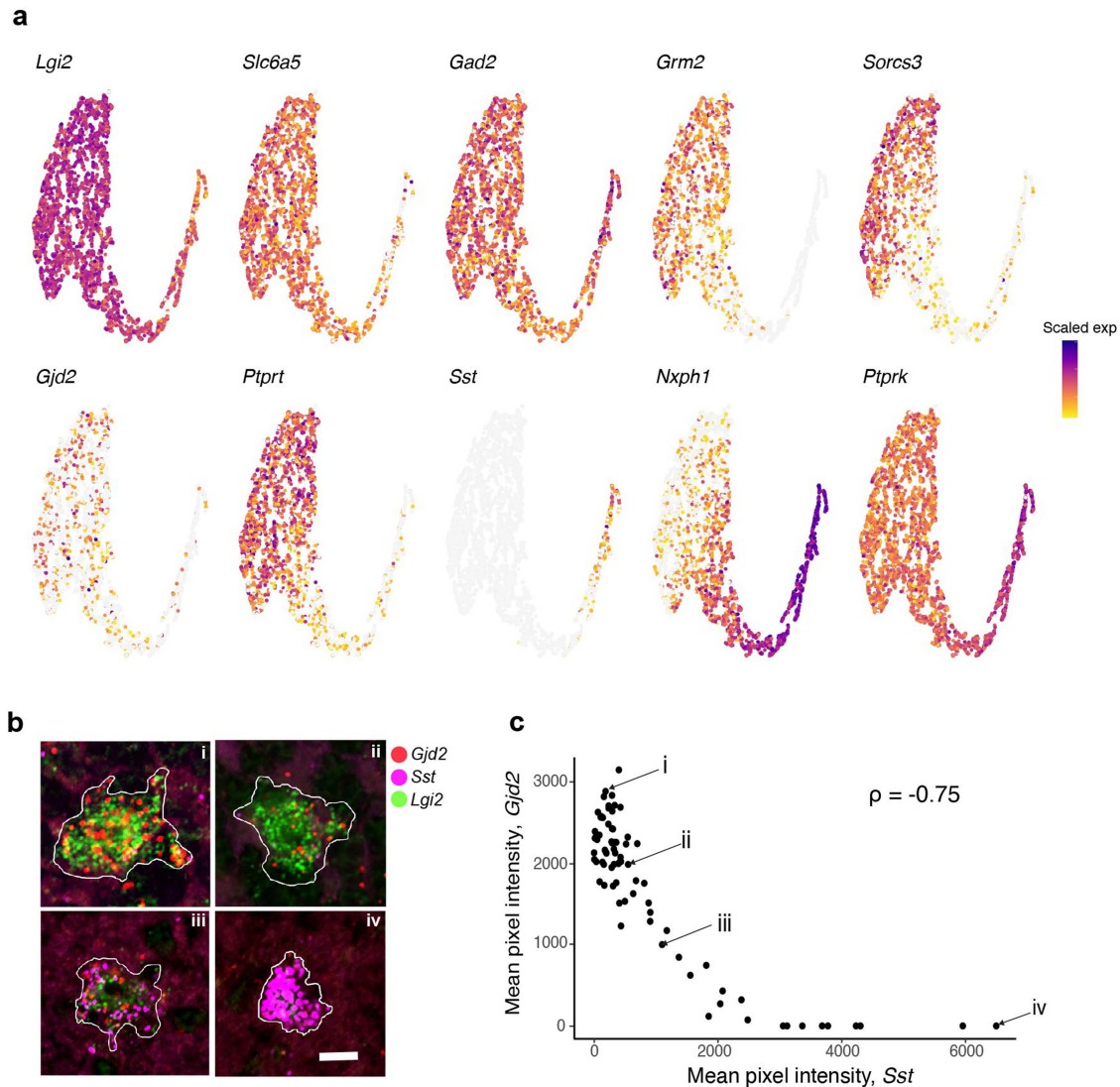

**Extended Data Fig. 9 | Molecular variation in Golgi interneuron clusters.**
**a**, UMAP visualization of expression of selected genes in the two Golgi interneuron clusters, showing variation in many of the same genes that define the MLI1/MLI2 distinction (*Sorcs3*, *Gjd2* and *Nxph1*). **b**, Four representative Golgi cells from an smFISH experiment for *Lgi2* (Golgi cell marker, green), *Sst* (Golgi_2 marker, purple), and *Gjd2* (higher in Golgi_1, red). Scale bar, 50 μm. *n* = 77. **c**, Quantification of mean smFISH pixel intensity for the genes *Sst* and *Gjd2*, across 77 cells. The four representative cells in **b** are labelled with the corresponding Roman numeral. Spearman correlation (ρ) is shown.

|---|---|

# Reporting Summary

Nature Research wishes to improve the reproducibility of the work that we publish. This form provides structure for consistency and transparency in reporting. For further information on Nature Research policies, see Authors & Referees and the Editorial Policy Checklist.

## Statistics

For all statistical analyses, confirm that the following items are present in the figure legend, table legend, main text, or Methods section.

| n/a | Confirmed | |
|---|---|---|
| ☐ | ☒ | The exact sample size (*n*) for each experimental group/condition, given as a discrete number and unit of measurement |
| ☐ | ☒ | A statement on whether measurements were taken from distinct samples or whether the same sample was measured repeatedly |
| ☐ | ☒ | The statistical test(s) used AND whether they are one- or two-sided<br>*Only common tests should be described solely by name; describe more complex techniques in the Methods section.* |
| ☐ | ☒ | A description of all covariates tested |
| ☐ | ☒ | A description of any assumptions or corrections, such as tests of normality and adjustment for multiple comparisons |
| ☐ | ☒ | A full description of the statistical parameters including central tendency (e.g. means) or other basic estimates (e.g. regression coefficient) AND variation (e.g. standard deviation) or associated estimates of uncertainty (e.g. confidence intervals) |
| ☐ | ☒ | For null hypothesis testing, the test statistic (e.g. *F*, *t*, *r*) with confidence intervals, effect sizes, degrees of freedom and *P* value noted<br>*Give P values as exact values whenever suitable.* |
| ☒ | ☐ | For Bayesian analysis, information on the choice of priors and Markov chain Monte Carlo settings |
| ☒ | ☐ | For hierarchical and complex designs, identification of the appropriate level for tests and full reporting of outcomes |
| ☒ | ☐ | Estimates of effect sizes (e.g. Cohen's *d*, Pearson's *r*), indicating how they were calculated |

*Our web collection on statistics for biologists contains articles on many of the points above.*

## Software and code

Policy information about availability of computer code

| Data collection | Mouse raw sequencing data was processed, aligned and converted to digital gene expression matrix format by CellRanger v.3.0.2 (available from 10x Genomics) with default settings. Human raw sequencing data was processed, aligned and converted to digital gene expression matrix format using the open-source Drop-seq alignment workflow. ISH imaging data was acquired with Nikon NIS Elements v4.6. Two-photon imaging data was acquired with Scanimage on MATLAB R2009b. Electrophysiology data was acquired with Igor Pro 8 (Wavemetrics) and software mafPC (https://www.xufriedman.org/mafpc). |
|---|---|
| Data analysis | Data was analyzed using Seurat v2.3.4, LIGER v0.4.2, Monocle v2.10.1, on R v3.5.3, and Monocle3 v0.2.2 on R v3.6.3.,in addition to custom code and scripts available at https://github.com/MacoskoLab/cerebellum-atlas-analysis. Two-photon images were processed with ImageJ v2.0.0-rc-68/1.53e, and ISH imaging data was processed with ImageJ v2.0.0-rc-68/1.53e and Nikon NIS AR. |

For manuscripts utilizing custom algorithms or software that are central to the research but not yet described in published literature, software must be made available to editors/reviewers. We strongly encourage code deposition in a community repository (e.g. GitHub). See the Nature Research guidelines for submitting code & software for further information.

## Data

Policy information about availability of data

All manuscripts must include a data availability statement. This statement should provide the following information, where applicable:

- Accession codes, unique identifiers, or web links for publicly available datasets
- A list of figures that have associated raw data
- A description of any restrictions on data availability

All processed data and annotations have been made freely available for download and visualization through an interactive Single Cell Portal study (https://singlecell.broadinstitute.org/single_cell/study/SCP795/). Raw and processed data that support the findings of this study have been deposited in GEO under accession number XXX and in at the Neuroscience Multi-omics (NeMO) Archive (https://nemoarchive.org/).

October 2018

# Field-specific reporting

Please select the one below that is the best fit for your research. If you are not sure, read the appropriate sections before making your selection.

☒ Life sciences ☐ Behavioural & social sciences ☐ Ecological, evolutionary & environmental sciences

For a reference copy of the document with all sections, see nature.com/documents/nr-reporting-summary-flat.pdf

# Life sciences study design

All studies must disclose on these points even when the disclosure is negative.

| | |
|---|---|
| Sample size | Sample size to achieve reliable sampling (probability > 10%) of even very rare cell types (~0.1% prevalence) was estimated from a prior study of mouse cerebellum (Saunders, Macosko et al, Cell 2018 and see Extended Data Fig. 1); multiple replicates for each sex were included to reduce the effects of inter-individual variation. For electrophysiology and cell imaging experiments, no statistical methods were used to predetermine sample size. Sample sizes were deemed sufficient due to stark differences in presence of spikelets as well as some basic electrical properties between MLI1 and MLI2, which aligned with an observed bimodal distribution of data. |
| Data exclusions | For transcriptomic data, pre-established exclusion criteria of a minimum of 500 UMI/nucleus were used, as this is a standard threshold used in previous studies. Nuclei with a lower number of UMIs were excluded from any analysis. |
| Replication | For electrophysiology and cell imaging experiments demonstrating physiological and gene expression differences between the MLI1 and MLI2 populations, experiments were performed independently in different mice (n=18 mice). When molecular identification after patch clamp was unsuccessful, 2 photon and electrophysiology data were included without a molecular label.<br><br>All other reported transcriptomic findings were consistent across biological replicates. |
| Randomization | No differential experimental treatments were applied to individuals or samples in this study. |
| Blinding | Blinding was not relevant in this study as no differential experimental treatments were applied. |

# Reporting for specific materials, systems and methods

We require information from authors about some types of materials, experimental systems and methods used in many studies. Here, indicate whether each material, system or method listed is relevant to your study. If you are not sure if a list item applies to your research, read the appropriate section before selecting a response.

## Materials & experimental systems

| n/a | Involved in the study |
|---|---|
| ☒ | Antibodies |
| ☒ | Eukaryotic cell lines |
| ☒ | Palaeontology |
| ☐ | ☒ Animals and other organisms |
| ☐ | ☒ Human research participants |
| ☒ | Clinical data |

## Methods

| n/a | Involved in the study |
|---|---|
| ☒ | ChIP-seq |
| ☒ | Flow cytometry |
| ☒ | MRI-based neuroimaging |

## Animals and other organisms

Policy information about studies involving animals; ARRIVE guidelines recommended for reporting animal research

| | |
|---|---|
| Laboratory animals | Mouse transcriptomic data was generated from 2 adult female and 4 adult male mice (60 days old; C57BL/6J, Jackson Labs). Mice were housed in a 12:12 light-dark cycle with ad libitum access to food and water. |
| Wild animals | No wild animals were used in the study. |
| Field-collected samples | No field-collected samples were used in the study. |
| Ethics oversight | All procedures involving animals at the Broad Institute were conducted in accordance with the US National Institutes of Health Guide for the Care and Use of Laboratory Animals under protocol number 0120-09-16. |

Note that full information on the approval of the study protocol must also be provided in the manuscript.

# Human research participants

Policy information about [studies involving human research participants](studies involving human research participants)

Population characteristics

*Describe the covariate-relevant population characteristics of the human research participants (e.g. age, gender, genotypic information, past and current diagnosis and treatment categories). If you filled out the behavioural & social sciences study design questions and have nothing to add here, write "See above."*

Recruitment

*Describe how participants were recruited. Outline any potential self-selection bias or other biases that may be present and how these are likely to impact results.*

Ethics oversight

Human cerebellum tissue assayed was obtained from the NIH Brain and Tissue Repository of California, through the NIH NeuroBioBank. The tissue was received without identifiable information, and did not meet the definition of human subjects research (project # NHSR-4235).

Note that full information on the approval of the study protocol must also be provided in the manuscript.

