## [Peer Review File · Nature]

Manuscript Title: A transcriptomic atlas of mouse cerebellar cortex reveals novel cell types

Editorial Notes:

Reviewer Comments & Author Rebuttals

Reviewer Reports on the Initial Version:

Referee #1 (Remarks to the Author):

Kozareva et al. present an atlas of the mouse cerebellum. The dataset focuses primarily on single-cell RNA-seq, but includes spatial, morphological, and functional characterization as well. In particular, they discover new subsets of molecular layer interneurons, and validate these differences with orthogonal assays and cross-species mapping. They conclude by highlighting the importance of examining multiple complementary modalities when defining cell states and ontologies.

I am not a neuroscientist, but I enjoyed the paper and found it to be of substantial conceptual interest. In particular, the manuscript has the following strengths

1. The manuscript contains a massive scRNA-seq dataset, but the manuscript is far more than the deposition of a large dataset. In particular, it provides a detailed and thoughtful discussion of discrete vs continuous states, and highlights the powerful (but limited) ability of transcriptomics to delineate between these possibilities. This is a conceptually important point that will resonate in many fields.
2. The authors perform extensive characterization of their newly discovered MLI heterogeneity. In particular, they show strong evidence of cross-species conservation, alongside functional analysis.
3. The ability to perform electrophysiological measurements alongside smFISH is exciting, and will be useful for many studies.

I have the following comments to improve the manuscript:

1. I did not find the 'cluster-connectivity' analysis (Fig. 1e) to be particularly informative. There isn't any follow-up on this in the manuscript, besides saying that some cluster distinctions vary more subtly than others, which is not surprising.
2. The identification of regional enrichment patterns amongst Purkinje cells is convincing, particularly with SLIDE-Seq validation. However, for non-specialists, the authors may wish to highlight why it is exciting to see regional specialization amongst these cells. Is there co-enrichment with other cell types that may suggest spatial restriction of different circuits?
3. I thought that the metric of continuity proposed by the authors was creative, but why was it applied to only $m=100$ variable genes?
4. It seems like the authors observe a continuum of spiking responses to UBC, with two extremes. They observe a similar phenomenon in the molecular data. However, it is not clear if the two continuum are related to each other. For example, are the authors certain that the ON_UBC state (extreme) does not correspond to UBC_2 (intermediate)? The parsimonious explanation is that the two patterns do match, but the manuscript would be strengthened if the authors presented evidence

of this for UBC.

5. It would be nice to see a heat map of all genes that are conserved DE between MLI_1 and MLI_2 in the human and mouse data, instead of just a couple of markers

6. It is interesting that MLI2 show extensive morphological heterogeneity, but little molecular heterogeneity. This is certainly possible, but is a surprising result. Do the authors see any conserved patterns of molecular heterogeneity between MLI1 and MLI2? I ask because there is shared morphological heterogeneity.

7. The authors state that MLI2 do not show evidence of gap junction coupling, but should the. MLI2 cluster than be definitively referred to as an MLI subtype? The molecular conservation with MLI_1 does not seem striking from Fig. 1 (a heat map would be beneficial here as well), and they seem to be more closely related to Purkinje-layer interneurons. This may be evident to a specialist, but was confusing to me.

Referee #2 (Remarks to the Author):

In this study, Kozavera et al. investigate the molecular and spatial diversity of cerebellar cortical neurons. A total of 611,034 high-quality nuclei were harvested from the different lobules of the cerebellum, individually sequenced and clustered in 46 cell types. The authors then characterize the spatial distribution of cell clusters and identify patterns of lobular distribution containing similar cell types. Select cross-cluster continuity and discreteness are addressed (using previously published approaches) together with corresponding relationships in electrophysiological properties

This manuscript constitutes the largest dataset of cerebellar cortical neurons to date, and, in addition to purely transcriptomic data, also provides select sm-fish and electrophysiological characterizations. In such, it is a useful and high-quality resource, although deep cerebellar nuclei, which are part of the cerebellum, are not covered here (title and appropriate sections should emphasize this focus on cortex). However, this remains an essentially descriptive study and the reported findings to not substantially alter our understanding of cerebellar organization. Thus, overall interest for a broad readership is probably limited.

I have the following comments:

- The authors made a great effort to dissect multiple lobules of the cerebellum but only superficially analyzed the spatial distribution of the differentially-expressed genes / cell types. It would be interesting to assess whether cell position is encoded in the transcriptome of a cell.

- The authors integrated single nuclei harvested from human tissue that confirm the presence of two types of molecular layer interneurons in human cerebellum. No other analysis of this potentially interesting human dataset is provided. It would be interesting to use this human data to compare genetic programs across the two species.

- Throughout the manuscript, and including in the abstract, the authors emphasize that their study highlights the importance of multi-modal integration to determine cell types in the brain. Nevertheless, the data presented in the manuscript do not originate from the systematic acquisition of data from different origins (i.e electrophysiology, smFish, morphology). Instead, the latter features are simply used to further characterize previously transcriptomically-segregated cell types. In this

sense, this is my view is a somewhat improper use of the term "multi-modal integration".

- In figure 4, two different panels are labeled "d".

- Line 147: the authors refer to figure 1b, but the previous sentence refers to figure 1c.

Referee #3 (Remarks to the Author):

Kozareva et al. report a cerebellar transcriptomic atlas and identify cellular and physiological diversity of neuron sub-types, particularly granule neurons, unipolar brush cells (UBCs) and molecular layer interneurons (MLIs). This paper has applied state of the art single cell transcriptomic approaches and limited in situ hybridisation validation and electrophysiology to characterise expression level and physiological differences between some cerebellar cell types. Major findings focus on molecular diversity within the UBC and MLI lineages.

General Critique:

The notion that regional or cellular diversity leads the ultimate architecture of the brain is clear and this paper adds to the list of studies showing such diversity. However, it fails to explain the upstream principle that determines diversity or show the functional significance of such diversity beyond ex vivo analysis. The findings are interesting in a specialist context and the transcriptomic approaches are rigorously performed. The electrophysiology of MLI subtypes and post sample cellular and molecular identification is elegant. However, quality and volume of in situ confirmation in the paper is limited and the phenotyping raises the question of what distinct functions result from the cellular and/or physiological differences described? No behavioural or other functional data is provided so that these findings remain preliminary. To raise the significance of the paper, the authors should go further to show how such diversity is developmentally specified and/or the in vivo relevance of UBC/MLI diversity using functional approaches.

Specific issues:

1. The single cell transcriptomic approach applied has resulted in the unsurprising finding of enhanced lineage diversity compared to that which was expected based on classic criteria. A systematic omission is the confirmation of differences at the protein level to substantiate conclusions.
2. Paragraph/line 156 describes distinct identities by gene expression and electrophysiology.
 - a. Is continuous variant UBC gene expression reflected at the protein level in keeping with electrophysiological responses? This should be shown for a set of marker genes.
 - b. Second, functional data are needed to show why such diversity is important in vivo, e.g., with genetic/functional studies.
 - c. The same functional considerations apply to the two MLI sub-types.
3. The findings show regional differences certain cell types but lacks mechanistic insight into how these are programmed. For example:
 - a. Granular neurons in posterior lobules of cerebellum showed greatest diversity (lines 96 through to 98). Why should this be the case? What is (are) the upstream principle(s) that would result in such regional specialisation? Providing this mechanism could greatly strengthen the paper.
 - b. The results show an interesting finding that there is a continuous or gradient pattern of UBC gene expression across regions, raising the question of what sets up this pattern? Is there a diffusible cue or activity gradient that is responsible? This is an important way the study could be extended into novel direction.
4. The stated conclusion that the study provides a "comprehensive cellular atlas" is in contrast with

the text's focus on neuron subtypes. The glial cells data should be better integrated. In the introduction there is a lack of integration of glial cells into description of cerebellar circuits; lines 101-103 are inconsistent with reports of astroglial heterogeneity. Also for intro, prior studies have used expression profiling during cerebellar development; some should be cited and discussed.

Author Rebuttals to Initial Comments:

We wish to thank the reviewers for their insightful comments, which have helped us improve the quality and clarity of our resubmitted manuscript. In response to their comments, we have made the following major changes:

1) We followed reviewer 3's suggestion to clarify lineage of the molecular layer interneuron types, collecting a total of 79,373 snRNA-seq profiles across E18, P0, P4, P8, P12, and P16 time points. Building a trajectory across these time points for the interneuron progenitors, we demonstrated that the MLI1 and MLI2 types begin to differentiate around P4, and complete their separation by P16. The differentiation occurs at the same time that the progenitors first enter the molecular layer (Fig. 4d and Extended Data Fig. 6). Interestingly, compared with MLI1 (or mature MLI2s), young MLI2 cells show very high *Fos* positivity, indicating differential and transient activity during developmental specification.

2) Reviewer 2 asked us to broaden the human and mouse cross-species analyses. We sampled an additional 63,636 profiles from another postmortem human donor, and performed cross-species analyses of all interneuron populations (UBC, Golgi, MLI/PLI, and granule). Our revised manuscript provides evidence for evolutionary conservation of the UBC molecular continuum in humans, and the conservation of the two Golgi clusters, as well as some of the granule clusters (Extended Data Fig. 4).

3) Finally, in response to comments by reviewers 1 and 3, we directly connected the molecular gradient in UBC expression with the observed continuum in electrophysiological responses. Specifically, because *Grm1* (mGluR1 receptor) and *Grm2* (mGluR2 receptor) are expressed in reciprocal gradients across UBCs, we used selective mGluR1 and mGluR2 agonists to show that mGluR1 and mGluR2 responses are graded across the UBC population (Fig. 3f, Extended Data Figure 5) with a significant number of cells that respond to both (Extended Data Figure 5, UBC7 and UBC9). This suggests the biphasic response profile likely corresponds to the molecular continuum defined by snRNA-seq.

In addition, we made the following minor changes to increase clarity and address minor issues in our methods:

1. We refined the color scale of our visualization function for lobule enrichment (used in Figure 2c,d,f,h, and Extended Data Figure 3d) to more consistently indicate extreme values in each plot.

2. We added genes to panel 3e to match expression staining images displayed in Extended Data Figure 3c.
3. We have modified the visualizations of UBCs in Figure 3c (top left) and 3d to better reflect the graded nature of their molecular expression, by using a colormap which indicates their pseudotime ordering.
4. We have updated our descriptions for the lobule enrichment analysis methods to improve overall clarity and transparency.

Referee #1:

Kozareva et al. present an atlas of the mouse cerebellum. The dataset focuses primarily on single-cell RNA-seq, but includes spatial, morphological, and functional characterization as well. In particular, they discover new subsets of molecular layer interneurons, and validate these differences with orthogonal assays and cross-species mapping.

They conclude by highlighting the importance of examining multiple complementary modalities when defining cell states and ontologies.

I am not a neuroscientist, but I enjoyed the paper and found it to be of substantial conceptual interest. In particular, the manuscript has the following strengths

1. *The manuscript contains a massive scRNA-seq dataset, but the manuscript is far more than the deposition of a large dataset. In particular, it provides a detailed and thoughtful discussion of discrete vs continuous states, and highlights the powerful (but limited) ability of transcriptomics to delineate between these possibilities. This is a conceptually important point that will resonate in many fields.*
2. *The authors perform extensive characterization of their newly discovered MLI heterogeneity. In particular, they show strong evidence of cross-species conservation, alongside functional analysis.*
3. *The ability to perform electrophysiological measurements alongside smFISH is exciting, and will be useful for many studies.*

I have the following comments to improve the manuscript:

1. *I did not find the 'cluster-connectivity' analysis (Fig. 1e) to be particularly informative. There isn't any follow-up on this in the manuscript, besides saying that some cluster distinctions vary more subtly than others, which is not surprising.*

Upon reflection, we definitely see the reviewer's point and have removed the panel in the revised manuscript.

2. *The identification of regional enrichment patterns amongst Purkinje cells is convincing, particularly with SLIDE-Seq validation. However, for non-specialists, the authors may wish to highlight why it is exciting to see regional specialization amongst these cells. Is there co-enrichment with other cell types that may suggest spatial restriction of different circuits?*

Part of the attraction of studying the cerebellum has been that it was thought to use a repeated circuit motif employing a small number of neuron types to perform

computations. Interest in the cerebellum has grown in recent years with the recognition that it contributes to an expanding list of diverse behaviors and neurological disorders. In parallel, evidence has accumulated that the complexity of the cerebellar circuit has been overly simplified and that the cerebellar cortex is specialized to perform different types of tasks. This study points to a repertoire of cell types and subtypes—larger than previously known—that allow specializations of the cerebellum within different regions that contribute to different computations. As the reviewer noted, the PCs were most clearly subspecialized within regions—most especially in posterior lobules. The UBCs are known to be particularly abundant in many of the same lobules; in addition, we did identify some GC types found selectively in posterior lobules, suggesting that there is greater local circuit heterogeneity than previously recognized.

We have revised the conclusion (reproduced below for the reviewer's convenience) to better emphasize the above points.

Here, we used high-throughput, region-specific transcriptome sampling to build a comprehensive taxonomy of cell types in the mouse cerebellar cortex, and quantify spatial variation across individual regions. Our joint analyses with postmortem human samples indicated that the mouse-defined neuronal populations were generally conserved in human (Extended Data Fig. 4), consistent with a recent comparative analysis in cerebral cortex³¹. We find considerably more regional specialization in PCs—especially in posterior lobules—than was previously recognized. These PC subtypes overlap with greater local abundances in UBCs and in distinct specializations in GCs, indicating a higher degree of regional circuit heterogeneity than previously thought. Our dataset is freely available to the neuroscience community^{32,33}, facilitating functional characterization of these populations, many of which are entirely novel.

One of the biggest challenges facing the comprehensive cell typing of the brain is the correspondence problem³⁴: how to integrate definitions of cell types based on the many modalities of measurement used to characterize brain cells. We found success by first defining populations using systematic molecular profiling, and then relating these populations to physiological and morphological features using targeted, joint analyses of individual cells. We were surprised that the cerebellar MLs—one of the first sets of neurons to be characterized over 130 years ago³⁵—are in fact composed of two molecularly and physiologically discrete populations, that each shows a similar morphological continuum along the depth axis of the ML. As comprehensive cell typing proceeds across other brain regions, we expect the emergence of similar basic discoveries that challenge and extend our understanding of cellular specialization in the nervous system.

3. I thought that the metric of continuity proposed by the authors was creative, but why was it applied to only $m=100$ variable genes?

The number of significant DEGs can vary substantially depending on the clusters being compared (for the MLI1 clusters, for example, we identified 326 significant DEGs, while for the Golgi/MLI1 clusters, we identified 1169 such genes). To make an apples-to-apples comparison, we used an equal number of genes for all comparisons--hence, the number needs to be less than or equal to the minimum number of DEGs for a comparison. Varying the numbers of genes used (within the above constraint), we observe only minimal differences:

In response to the reviewer's comment and to improve comprehensiveness, we have updated the method description and Fig. 3b to include 200 genes instead.

4. It seems like the authors observe a continuum of spiking responses to UBC, with two extremes. They observe a similar phenomenon in the molecular data. However, it is not clear if the two continua are related to each other. For example, are the authors certain that the ON_UBC state (extreme) does not correspond to UBC_2 (intermediate)? The parsimonious explanation is that the two patterns do match, but the manuscript would be strengthened if the authors presented evidence of this for UBC.

We performed additional pharmacology experiments to directly connect the two continua, and have included these results in Fig. 3f and a new Extended Data Fig. 5 (reproduced below). Specifically, expression of *Grm1* (mGluR1 receptor) and *Grm2* (mGluR2 receptor) is anticorrelated across the UBC molecular continuum. Pressure applications of selective agonists for mGluR1 and mGluR2 in conjunction with glutamate in slice experiments allowed us to relate the ON spiking phenotype with a high mGluR1 response and low mGluR2 response to the *Grm1*+ end of the molecular continuum (Extended Data Fig. 5, UBC16). Both mGluR1 and mGluR2 responses were graded across the UBC population (Fig. 3f, reproduced below) and there were a significant number of cells which responded to both (Extended Data Fig. 5, UBC7 and UBC9). This suggests the biphasic response profiles of these cells likely correspond to cells in the middle of the

molecular continuum.

Figure 3f and Extended Data Figure 5: UBCs exhibited graded synaptic response to glutamate, mGluR1 and mGluR2 agonists.

Left, top: Schematic describing whole-cell recordings obtained from UBCs evoked by pressure application of glutamate (left, black), the mGluR1 agonist DHPG (middle, red) and the mGluR2 agonist LY354740 (right, blue) using three pipettes placed within 20 μm of the recorded cell. Left, bottom: Responses of four representative UBCs are shown in order of excitatory to inhibitory responses (top to bottom). These experiments are designed to determine if the presence of mGluR1 and mGluR2 account for the glutamate-evoked responses. They are well suited to quantifying the magnitude of responses, but the time courses of responses evoked by selective agonists will tend to be slowed because uptake systems only reduce glutamate levels in the slice, and are not effective at reducing the levels of artificial agonists. As shown in the representative UBCs, cells where glutamate evoked primarily an inward current (UBC16), there was a very large mGluR1 component and a tiny mGluR2 component. The opposite was true for UBCs where glutamate evoked primarily an outward current (UBC2). For intermediate cells such as UBC9 and UBC7, mGluR1 and mGluR2 components were both prominent. Evoked currents from each application are summarized in the correspondingly colored plots in the right.

5. It would be nice to see a heat map of all genes that are conserved DE between MLI_1 and MLI_2 in the human and mouse data, instead of just a couple of markers

We appreciate this suggestion and now include a heatmap showing multiple markers conserved between the MLI1 and MLI2 populations across mouse and human profiles in Extended Data Fig. 4d:

6. It is interesting that MLI2 show extensive morphological heterogeneity, but little molecular heterogeneity. This is certainly possible, but is a surprising result. Do the authors see any conserved patterns of molecular heterogeneity between MLI1 and MLI2? I ask because there is shared morphological heterogeneity.

Like the reviewer, we were very interested in whether MLI2s displayed internal molecular variation. The MLI1 molecular variation is spatially patterned, with MLI1_1, marked by *Grm8*, localizing more to the inner third of ML (where basket cells are more abundant), and MLI1_2, marked by *Npas3*, localizing more to the outer third (where stellate cells are more abundant). We did identify an axis of variation within MLI2, marked by several genes, including *Clmp*, *Sgcz*, *Tenm1*, and *Grin2a*, which also appears differentially expressed between the MLI1_1 and MLI1_2 clusters (as shown in the figure below).

However, smFISH for these genes failed to show spatial gradients among the MLI2 cells (as we had previously been able to do with *Grm8* in the MLI1 cells). As one can see in the feature plots above, most of the genes vary quite subtly within MLI2 (more subtly than the other distinctions we validated in this manuscript with smFISH), so it's possible that this variation is just technical noise. Alternatively, this variation could be present in MLI2 but simply not correlated with spatial position (or morphology). We hope to continue investigating the molecular correlates of the MLI2 population's morphological heterogeneity in future work.

7. The authors state that MLI2 do not show evidence of gap junction coupling, but should the. MLI2 cluster than be definitively referred to as an MLI subtype? The molecular conservation with MLI_1 does not seem striking from Fig. 1 (a heat map would be beneficial here as well), and they seem to be more closely related to Purkinje-layer interneurons. This may be evident to a specialist, but was confusing to me.

Indeed, as seen in the dot plot in Fig 4a, many of the genes that distinguish MLI2 from MLI1 are genes that are also expressed in Purkinje-layer interneurons (for example, the marker *Nxph1* itself). Here, the developmental data we gathered for our revised manuscript turned out to be quite informative. Specifically, we gathered cerebellar profiles every four days from postnatal days zero to 16, and used Monocle3 to build a pseudotime trajectory based upon gene expression. The MLI populations clearly branch from each other much later in development than from the PLIs (which can be identified by their expression of the canonical marker *Kih1*):

This analysis, combined with the dendrogram of the adult data that positions the MLI2s next to the MLI1s, supports the idea that the MLIs are more closely related to each other than MLI2s are to PLIs.

Referee #2:

In this study, Kozavera et al. investigate the molecular and spatial diversity of cerebellar cortical neurons. A total of 611,034 high-quality nuclei were harvested from the different lobules of the cerebellum, individually sequenced and clustered in 46 cell types. The authors then characterize the spatial distribution of cell clusters and identify patterns of lobular distribution containing similar cell types. Select cross-cluster continuity and discreteness are addressed (using previously published approaches) together with corresponding relationships in electrophysiological properties

Though it builds on prior work to construct trajectories from single cell data (most especially in developmental contexts), our current work does in fact contribute a novel approach to assessing cluster discreteness and continuity. Specifically, our method fits a logistic curve to each gene's expression across the trajectory, allowing us to quantify, for each gene (using the maximum slope of the fitted curve, termed *m*), its degree of discreteness or continuity. We can then assess, across a representative set of differentially expressed genes, the degree of continuity of the clusters (Fig 3b). We find this approach to be quite robust and comparable across different cell types and datasets.

We should note that this sort of quantitative approach will be increasingly important in the comparative analysis of cell types across the brain. There have been several

cases of single cell analysis defining discrete populations across, for example, two different regions (e.g., see PMID 30382198) that actually form two ends of a more continuous axis of variation when intervening regions are included in analysis.

This manuscript constitutes the largest dataset of cerebellar cortical neurons to date, and, in addition to purely transcriptomic data, also provides select sm-fish and electrophysiological characterizations. In such, it is a useful and high-quality resource, although deep cerebellar nuclei, which are part of the cerebellum, are not covered here (title and appropriate sections should emphasize this focus on cortex).

We apologize for any confusion, and have changed the title and relevant sections to make explicit that our focus is on the cerebellar cortex.

However, this remains an essentially descriptive study and the reported findings do not substantially alter our understanding of cerebellar organization. Thus, overall interest for a broad readership is probably limited.

The ability to relate circuitry to neural processing has long made the cerebellum of broad general interest to neuroscientists. Part of the attraction has been that the cerebellar cortex was thought to use a repeated circuit motif employing a small number of neuron types to perform computations. Interest in the cerebellum has grown in recent years with the recognition that it contributes to an expanding list of diverse behaviors and neurological disorders. In parallel, evidence has accumulated that the complexity of the cerebellar circuit has been overly simplified and that the cerebellar cortex is specialized to perform different types of tasks. This study clarifies the circuit elements of the cerebellum, and points to a repertoire of cell types and subtypes--larger than previously known--that allow specializations of the cerebellum within different regions that contribute to different computations.

Five specific advances emerge from this work:

1. The single most remarkable finding is that molecular layer interneurons actually consist of two different types. Previously, there was no suspicion that there were two intermingled populations of MLIs throughout the molecular layer, only one of which is gap junction coupled. Functionally, we found MLI2s have low initial firing rates but are highly excitable, making them responsive to elevated granule cell activity and therefore a putative gain control mechanism. The gap junction coupling of MLI1 will tend to make these MLIs fire synchronously and provide precisely timed inhibition to PCs, suited to regulating the timing of pauses and firing of PCs. We hypothesize that MLI1 regulates the timing of PC firing and gain control, while MLI2 regulates the firing rate of PCs. Functional *in vivo* studies are needed to test this hypothesis, but will require higher temporal resolution than can be provided by calcium indicators and are beyond the scope of our study.

In our revised manuscript, we build on the MLI results, identifying the developmental time window when these populations separate, and discover the transient, MLI2-specific expression of *Fos* at the time of subtype specification. We have modified Fig. 4 to include these new findings (panels d and e, reproduced below):

2. We have found that UBCs constitute a single population with continuous varying molecular properties rather than consisting of a small number of discrete subtypes. Our functional studies suggest that this has important implications for temporal processing. The population of UBCs exhibit a graded continuum of temporal responses that are consistent with the molecular properties of UBCs. Many cell types in other brain regions exhibit a similar continuum of molecular properties, making this finding of general interest. In the revised manuscript, we include pharmacology experiments to directly relate the functional and molecular continua, and modify the presentation of data in Figure 3 to better reflect the continuum of molecular properties of UBCs.
3. We have identified subtypes of Purkinje cells (PCs) and granule cells (GCs) and regional differences in the presence of these subtypes. Previous studies have hinted at functional and molecular heterogeneity in PCs and GCs (e.x. PMIDs 32022688, 17151600 and 16818382), but we find that the degree of sub-specialization is far more than previously known. For the PCs, we identify 9 subpopulations, 7 of which are concentrated in the posterior lobules. For the GCs, we find five subsets: one concentrated in the anterior lobules, another in the posterior lobules, a third restricted only to region X, and two additional distinct clusters that constitute minority populations within all lobules.
4. Our comprehensive approach allowed us to identify all cell types and subtypes of the cerebellar cortex. This is an important step in clarifying cerebellar circuitry that promises to contribute to a breakthrough in determining the categories of cells and ultimately in determining the function of these cell types. This includes the first molecular characterization of relatively rare types

that are part of the Purkinje layer interneuron category. These interneurons are distinct from molecular layer interneurons and in all likelihood include the Lugaro cells, globular cells and candelabrum cells (PMID 8300903, 12421603, 17099896, 22235322), which are all cell types that have not been characterized at all at the molecular and functional levels. Future studies will build on the molecular information we have provided on PLIs to perform such molecular and functional studies. Indeed, we are in the process of using this information to perform a comprehensive characterization of candelabrum cells, a cell that has only been described anatomically.

This is an example of the type of study that will build on our findings. We have also provided the first molecular characterization of Golgi cells (PMID 17099896), and show differential expression of the gap junction subunit *Gjd2* (Extended Data Fig. 9). This will be an important step in clarifying the subtypes of these enigmatic cells that control the excitability of the granular layer.

5. In the revised manuscript, we extend our human-mouse analyses to show that the distinctions we identified by profiling mice are largely conserved (Extended Data Fig. 4). This includes the continuously varying UBC properties, and the distinct MLI1 and MLI2 categories. The details of these new analyses are below, where reviewer 2 specifically commented about our human-mouse integrative analysis.

We would like to also provide some additional clarification about just how quantitatively different the MLI1 and MLI2 populations actually are, relative to other recent discoveries by single cell analysis. Using our discreteness/continuity analysis that we deployed on the UBCs in Figure 3, we compared the discreteness of the MLI1 vs. MLI2 to: 1) other distinctions discovered by a recent single-cell study (Tasic et al., PMID 30382198, left); and 2) canonical pairs of cell types with very well established functional cell type differences from another recent single cell study (PMID 30096299) and the Tasic et al. study (chandelier and basket cells in the motor cortex).

The MLI1/MLI2 distinction is far larger than what was discovered in the recent single-cell study, and is much more similar to canonical differences amongst the cell types well-studied by neuroscientists (indirect versus direct SPNs in striatum, or CA1 versus CA3 pyramidal neurons in hippocampus).

I have the following comments:

- The authors made a great effort to dissect multiple lobules of the cerebellum but only superficially analyzed the spatial distribution of the differentially-expressed genes / cell types. It would be interesting to assess whether cell position is encoded in the transcriptome of a cell.

We feel that the spatial information of differentially expressed genes is one of the strengths of our study. We strived to provide an in-depth analysis by systematically analyzing the distribution of each cell cluster across each lobule (Fig. 2), and identifying many spatially discrete molecular specializations (almost all novel) in the core cell types of the cerebellum--the Purkinje cells and the granule cells--along with Bergmann glia. Our work provides directly testable hypotheses for cerebellar biologists about how molecular specialization relates to functional specialization of specific cerebellar regions.

The reviewer poses an interesting question about whether some genes may specify lobular position across multiple cell types. To explore this question, we looked for genes whose expression was better predicted by cell location than knowledge of cell type alone. To find the genes which consistently had this property across multiple replicates, we used logistic regression models to compare each variable's marginal predictive power. For each gene, we compared the predictive power gained by going from a model with only cell type information to a model with both cell type *and* lobular location information, adjusting for the gene's mean expression and the cells' donor identity.

We found only 6 genes (see table below) that had the desired property of consistently gaining more marginal predictive power from lobular information than

from cell type information (i.e. positive percentage in the second to last column). Upon examination, all of these genes are clearly technical artifacts stemming from non-neuronal cell cross-contamination. For example, *Ttr* is extremely highly expressed in the choroid plexus and, during preparation of nuclei for sequencing, can leak into profiles of cells from regions that flank the fourth ventricle, where choroid plexus is situated (e.g. region IX).

While we did not find evidence for position-encoding genes in this study, we plan to obtain finer-grained cellular location data in future studies, which could help give greater insight into this intriguing question.

Gene Name	Mean Percent of Total Deviance Reduction from Cell Type	Mean Percent of Total Deviance Reduction from Lobular Regions	Difference of Previous Two Columns	Putative Explanation
Ttr	3.63%	96.37%	92.74%	Choroid plexus contamination
Gm42418	15.65%	84.35%	68.69%	High choroid expression
En1	18.89%	81.11%	62.23%	Choroid plexus contamination
Ptgds	34.05%	65.95%	31.89%	Endothelial Cell
1500002C15Rik	36.34%	63.66%	27.31%	Fibroblast Cell
Gm17275	45.68%	54.32%	8.65%	High choroid expression

- The authors integrated single nuclei harvested from human tissue that confirm the presence of two types of molecular layer interneurons in human cerebellum. No other analysis of this potentially interesting human dataset is provided. It would be interesting to use this human data to compare genetic programs across the two species.

At the reviewer's suggestion, we added another postmortem donor to increase our human dataset by 63,636 profiles. We used LIGER to generate integrative analyses of all cerebellar interneuron populations: MLI/PLIs, Golgis, UBCs, and granule cells. In humans, we observe clear conservation of: 1) the MLI1/MLI2 split; 2) the UBC continuum; 3) the two Golgi subpopulations; and 4) the lobule X-specific granule interneuron population marked by expression of the gene *Galnt16*. There is also weaker evidence of the *Chrm3+* granule cluster being present in human granule cells. These results are consistent with another recent cross-species analysis of cortex, which found that cell types there are largely conserved between human and mouse (PMID 31435019).

Extended Data Figure 4: Integrative analysis of human and mouse cerebellar interneuron profiles. UMAP representation of the integrative analyses of UBC (1,613 mouse; 3,893 human) (a), MLI/PLI (45,555 mouse; 14,971 human) (c), Golgi (3,989 mouse; 1,059 human) (e), and granule (119,972 mouse; 130,335 human) (g) cells, colored by species (top), or joint cluster (bottom, for MLI/PLI, Golgi, and granule only). Heat maps showing expression of selected genes in UBC (b), MLI/PLI (d), and Golgi (f). Profiles are segregated both by species and cluster. h, Left, dot plot showing expression of selected genes in granule clusters, within human (red) and mouse (blue). Right, proportional representation of lobule dissections across the granule clusters. Granule cluster numbers approximately correspond to the mouse-only clusters shown in Fig. 2e.

- Throughout the manuscript, and including in the abstract, the authors emphasize that their study highlights the importance of multi-modal integration to determine cell types in the brain. Nevertheless, the data presented in the manuscript do not originate from the systematic acquisition of data from different origins (i.e. electrophysiology, smFish, morphology). Instead, the latter features are simply used to further characterize previously transcriptomically-segregated cell types. In this sense, this is my view is a somewhat improper use of the term “multi-modal integration”.

We apologize that the language we used could give readers the impression that we conducted a systematic, unbiased multi-modal analysis. In fact, our point was exactly the opposite—that by assembling a systematic and unbiased molecular definition of cell types, and using that RNA-seq data to contextualize and drive follow-up morphological and physiological experiments, we were able to provide integrative definitions of cell types—such as MLIs and UBCs—whose properties had not been well understood previously. This is a framework we envision deploying for cell typing across the rest of the mammalian brain. In the revised manuscript, we more clearly articulate that our proposed framework begins with systematic molecular sampling. The relevant lines from the revised conclusion below are reproduced for the reviewer's convenience.

One of the biggest challenges facing the comprehensive cell typing of the brain is the correspondence problem²⁹: how to integrate definitions of cell types based on the many modalities of measurement used to characterize brain cells. We found success by first defining populations using systematic molecular profiling, and then relating these populations to physiological and morphological features using targeted, joint analyses of individual cells. We were surprised to find that the cerebellar MLIs—one of the first sets of neurons to be characterized over 130 years ago³⁰—are in fact composed of two molecularly and physiologically discrete populations, that each themselves show a similar morphological continuum along the depth axis of the ML. As comprehensive cell typing proceeds across other brain regions, we expect the emergence of similar basic discoveries that challenge and extend our understanding of cellular specialization in the nervous system.

- In figure 4, two different panels are labeled “d”.

We apologize for this error, and now label the panels appropriately.

- Line 147: the authors refer to figure 1b, but the previous sentence refers to figure 1c.

We thank the reviewer for identifying this error, and have corrected it.

Referee #3:

Kozareva et al. report a cerebellar transcriptomic atlas and identify cellular and physiological diversity of neuron sub-types, particularly granule neurons, unipolar brush cells (UBCs) and molecular layer interneurons (MLIs). This paper has applied state of the art single cell transcriptomic approaches and limited in situ hybridisation validation and electrophysiology to characterise expression level and physiological differences between some cerebellar cell types. Major findings focus on molecular diversity within the UBC and MLI lineages.

General Critique:

The notion that regional or cellular diversity leads the ultimate architecture of the brain is clear and this paper adds to the list of studies showing such diversity. However, it fails to explain the upstream principle that determines diversity or show the functional significance of such diversity beyond ex vivo analysis.

The findings are interesting in a specialist context and the transcriptomic approaches are rigorously performed. The electrophysiology of MLI subtypes and post sample cellular and molecular identification is elegant. However, quality and volume of in situ confirmation in the paper is limited and the phenotyping raises the question of what distinct functions result from the cellular and/or physiological differences described? No behavioural or other functional data is provided so that these findings remain preliminary. To raise the significance of the paper, the authors should go further to show how such diversity is developmentally specified and/or the in vivo relevance of UBC/MLI diversity using functional approaches.

We took the reviewer's suggestion and profiled the cerebellum during the developmental window when the molecular layer interneurons are specified: starting at E18, and sampling at P0, P4, P8, P12, and P16. Using these additional profiles, we identified the developmental time window when the MLI1 and MLI2 populations separate (P4-P16), and postulate, because of asymmetric expression of *Fos*, that the subtype specification may be driven by differential activity (Fig. 4d,e and Extended Data Fig. 6; see also the discussion of these results in the next response section).

Specific issues:

1. The single cell transcriptomic approach applied has resulted in the unsurprising finding of enhanced lineage diversity compared to that which was expected based on classic criteria.

We respectfully disagree that the analysis of our dataset resulted in "unsurprising" findings about the structure and function of cell types in the cerebellum. The ability to relate circuitry to neural processing has long made the cerebellum of broad general interest to neuroscientists. Part of the attraction has been that the cerebellar cortex was thought to use a repeated circuit motif employing a small number of neuron types to perform computations. Interest in the cerebellum has grown in recent years with the recognition that it contributes to an expanding list of diverse behaviors and neurological disorders. In parallel, evidence has accumulated that the complexity of the cerebellar circuit has been overly simplified and that the cerebellar cortex is specialized to perform different types of tasks. This study clarifies the circuit elements of the cerebellum, and points to a repertoire of cell types and subtypes--larger than previously known--that allow specializations of the cerebellum within different regions that contribute to different computations.

The single most remarkable finding is that molecular layer interneurons actually consist of two different types. Previously, there was no suspicion that there were two intermingled populations of MLIs throughout the molecular layer, only one of which is gap junction coupled. Functionally, we found MLI2s have low initial firing rates but are highly excitable, making them responsive to elevated granule cell activity and therefore a putative gain control mechanism. The gap junction coupling of MLI1 cells will tend to make these MLIs fire synchronously and provide precisely timed inhibition to PCs, suited to regulating the timing of pauses and firing of PCs.

These studies suggest the hypothesis that MLI1 regulates the timing of PC firing and gain control, while MLI2 regulates the firing rate of PCs. Functional *in vivo* studies are needed to test this hypothesis.

In our revised manuscript, we build on the MLI results, identifying the developmental time window when these populations separate, and discover the transient, MLI2-specific expression of *Fos* at the time of subtype specification. We have modified Fig. 4 to include these new findings (panels d and e, reproduced below):

We would like to also provide some additional clarification about just how quantitatively different the MLI1 and MLI2 populations actually are, relative to other recent discoveries by single cell analysis in the brain. Using our discrete/continuous analysis that we deployed on the UBCs in Fig. 3, we compared the discreteness of the MLI1 vs. MLI2 to: 1) other distinctions discovered by a recent single-cell study (Tasic et al., PMID 30382198, left); and 2) canonical pairs of cell types with very well established functional differences that were sampled in another recent single cell study (Saunders et al., PMID 30096299) and the Tasic et al. study (chandelier and basket cells in the motor cortex). The MLI1/MLI2 distinction is far larger than what was discovered in the recent single-cell study, and is much more similar to canonical differences amongst the cell types well-studied by neuroscientists (indirect versus direct SPNs in striatum, or CA1 versus CA3 pyramidal neurons in hippocampus).

A systematic omission is the confirmation of differences at the protein level to substantiate conclusions.

2. Paragraph/line 156 describes distinct identities by gene expression and electrophysiology.

a. Is continuous variant UBC gene expression reflected at the protein level in keeping with electrophysiological responses? This should be shown for a set of marker genes.

We agree that gene expression differences, on their own, can sometimes prove difficult to interpret from a functional perspective (given that proteins of course are the functional units). This is precisely why we were so excited by the electrophysiological results we presented in Fig. 4: the binary gene expression difference we observed in *Gjd2* was exactly mirrored by the spikelet recordings (whose differential behavior is of course a direct observation of differences in functional GJD2 protein expression).

In our revised manuscript, we extend our functional validation, including pharmacological studies of UBCs using mGluR1 and mGluR2 inhibitors, directly connecting the transcriptional continuum we identified in UBCs with the (protein-driven) functional continuum that we characterized in electrophysiological recordings. These new findings are presented in Fig. 3f and Extended Data Fig. 5. They establish that glutamate application evokes responses in UBCs that reflect continuously and inversely graded mGluR1 and mGluR2 mediated currents. These functional studies are consistent with the molecular properties of UBCs.

Figure 3f and Extended Data Figure 5: UBCs exhibited graded synaptic response to glutamate, mGluR1 and mGluR2 agonists.

Left, top: Schematic describing whole-cell recordings obtained from UBCs evoked by pressure application of glutamate (left, black), the mGluR1 agonist DHPG (middle, red) and the mGluR2 agonist LY354740 (right, blue) using three pipettes placed within 20 μm of the recorded cell. Left, bottom: Responses of four representative UBCs are shown in order of excitatory to inhibitory responses (top to bottom). These experiments are designed to determine if the presence of mGluR1 and mGluR2 account for the glutamate-evoked responses. They are well suited to quantifying the magnitude of responses, but the time courses of responses evoked by selective agonists will tend to be slowed because uptake systems only reduce glutamate levels in the slice, and are not effective at reducing the levels of artificial agonists. As shown in the representative UBCs, cells where glutamate evoked primarily an inward current (UBC16), there was a very large mGluR1 component and a tiny mGluR2 component. The opposite was true for UBCs where glutamate evoked primarily an outward current (UBC2). For intermediate cells such as UBC9 and UBC7, mGluR1 and mGluR2 components were both prominent. Evoked currents from each application are summarized in the correspondingly colored plots in the right.

b. Second, functional data are needed to show why such diversity is important in vivo, e.g., with genetic/functional studies.

c. The same functional considerations apply to the two MLI sub-types.

We wholeheartedly agree with the reviewer that these suggestions are important and likely to lead to some intriguing findings about cerebellar function. Our primary goal in the present study is to report a comprehensive inventory of cerebellar cortex cell types, to show how this inventory challenges our existing understanding of cerebellar circuitry, and to justify that these transcriptionally defined types are likely to have significant functional consequences. We have supported the latter assertion

in the following specific ways:

- 1) Elucidation of a molecular continuum within UBCs that explains the marked heterogeneity in electrophysiological responses observed in response to applied glutamate. We connected these two continua directly using pharmacological inhibitors of the differentially expressed mGluR1 and mGluR2 receptors.
- 2) Identification of numerous electrophysiological properties that distinguish the two MLI subtypes, including the demonstration of differential gap junction properties between the two types, consistent with their differential expression of *Gjd2* (a.k.a. connexin 36).

Given the substantial technical challenges associated with recording from these populations, *in vivo* characterization of these populations is well outside the scope of the current study, but is certainly work that we hope to pursue in the future.

3. The findings show regional differences certain cell types but lacks mechanistic insight into how these are programmed. For example:

a. Granular neurons in posterior lobules of cerebellum showed greatest diversity (lines 96 through to 98). Why should this be the case? What is (are) the upstream principle(s) that would result in such regional specialisation? Providing this mechanism could greatly strengthen the paper.

Rather than being posteriorly enriched, Fig. 2f shows that three of the five granule subpopulations are spatially enriched--one in the anterior lobules, a second in the lateral posterior lobules, and the third in the nodulus. Consistent with this, in lines 96-98, we state:

We also observed regional specialization in excitatory interneurons and Bergmann glia. Among the 5 GC subtypes (Fig. 2e), three had significant and cohesive spatial enrichment patterns (subtypes 1, 2, and 3, Fig. 2f, Extended Data Fig. 3c).

In our analysis of Purkinje neurons in Fig 2d, we show that there is more molecular diversity in the posterior lobules than in the anterior lobules. The higher degree of Purkinje molecular diversity there is correlated with greater known diversity in connectivity: for example, some Purkinje cells located in posterior lobules bypass the deep cerebellar nuclei and synapse directly onto targets in the medulla (PMID 7852634) and pons (PMIDs 10678528 and 29467628). We hope that our cerebellar atlas inspires future work in the direction the reviewer suggests--to use our molecular data to build targeted transgenic tools (which take years to develop) to access these previously unknown Purkinje types, and interrogate them functionally and developmentally.

b. The results show an interesting finding that there is a continuous or gradient pattern of

UBC gene expression across regions, raising the question of what sets up this pattern? Is there a diffusible cue or activity gradient that is responsible? This is an important way the study could be extended into novel direction.

In the present work, we have not found that the continuous molecular variation in UBCs is spatially patterned. We have revised our descriptions of the UBCs as we realized the misunderstanding expressed by the reviewer likely stems from our use of the word “gradient,” which is often interpreted to have a spatial meaning. What we have shown here is that individual UBCs in the cerebellar cortex vary continuously in their gene expression, and it specifies a continuum of electrophysiological properties in response to glutamate application. In the revised manuscript, we include new experiments to make the connection between these molecular and functional continua explicit with our electrophysiological recordings of UBCs treated with selective mGluR1 and mGluR2 agonists (Fig. 3f, Extended Data Fig. 5).

4. The stated conclusion that the study provides a “comprehensive cellular atlas” is in contrast with the text’s focus on neuron subtypes. The glial cells data should be better integrated. In the introduction there is a lack of integration of glial cells into description of cerebellar circuits; lines 101-103 are inconsistent with reports of astroglial heterogeneity.

We emphasized the neuronal diversity in our study because of its clear biological novelty; our analysis of glial populations recapitulated the findings of previous studies (with the exception of the Bergman glia--see below). Specifically, two recent broad surveys of cell types across the mouse brain (PMIDs 30096299 + 30096314) both found minimal regional specialization of glial populations compared with neurons. While non-telencephalic astrocytes could be distinguished from telencephalic astrocytes (using the marker *Agt*), the oligodendrocytes, microglia, and endothelial populations surveyed in those studies were all very similar across brain regions.

To demonstrate that our study supports this conclusion, we performed an integrated analysis with our cerebellum dataset together with profiles from the mouse primary motor cortex (*Biorxiv* 2020.02.29.970558). Upon inspection of the UMAP embedding of this analysis, it is clear that some populations grossly separate by region, while others are intermixed:

When we clustered this analysis and traced the contribution of each joint cluster to the annotated clusters of the original datasets, we find that the shared clusters are glial populations (highlighted in bold in the figure below), while the neuronal populations are dataset-specific (and hence specific to each region):

We do see dataset-specific populations of astrocytes, consistent with the aforementioned work showing differences between telencephalic and non-telencephalic astrocytes. There is also a cortex-specific population of leptomenigeal cells (VLMC), which simply represents a difference in dissection (our study removed the overlying meninges). In addition, as expected, the cerebellar Bergmann glia (BG) are quite specialized, having no corresponding population in the cerebral cortex. We highlighted BG transcriptional specializations, even identifying lobule-to-lobule variation amongst them (lines 96-99). We are not aware of any other reported instances of astroglial variation within anatomical subregions of a structure (for example, across regions of cortex).

We do appreciate the reviewer's point about the important role glia play in establishing and maintaining cerebellar neuronal circuitry and now include mention of the Bergmann glia in the introduction (line 36).

Also for intro, prior studies have used expression profiling during cerebellar development; some should be cited and discussed.

We agree with the reviewer and have cited these developmental datasets in the introduction (lines 48-50).

Reviewer Reports on the First Revision:

Referee #1 (Remarks to the Author):

I appreciate the authors carefully considering all of my suggestions, and collecting new and convincing experimental data in response. I strongly support publication of the revised manuscript.

Referee #2 (Remarks to the Author):

We thank the authors for their extensive revisions, which address my concerns. The manuscript is significantly enhanced and in my view suitable for publication

Referee #3 (Remarks to the Author):

In the revised manuscript, the authors have augmented data for cerebellar developmental time course, human conservation and analytical conclusions, and they have included a new glutamate receptor agonist experiment. However, they have responded rather selectively to prior comments and major issues raised in the first round of review were not addressed, such as how diversity of neuron subtypes is established and/or reflected by their unique functions in vivo. It is possible that

the authors misinterpreted prior suggestions. For example, while the developmental time course captures the timing of diversification, it fails to identify the regulatory pathway that determines divergence of interneuron cell fate. The in vivo significance of the continuous UBC gradient is not tested. Thus, the paper remains of higher interest to a specialist journal.

Author Rebuttals to First Revision:

N/A